



**Evaluation of six geothermal heat flux maps for the Antarctic Lambert-Amery**
**glacial system**
Haoran Kang[1],Liyun Zhao[1,2*], Michael Wolovick[1,5], John C. Moore[1,3,4*]
[1] College of Global Change and Earth System Science, Beijing Normal University,
Beijing 100875, China
[2] Southern Marine Science and Engineering Guangdong Laboratory (Zhuhai),
China
[3] CAS Center for Excellence in Tibetan Plateau Earth Sciences, Beijing 100101,
China
[4] Arctic Centre, University of Lapland, Rovaniemi, Finland
[5] Alfred Wegener Institute, Bremerhaven, Germany
* Corresponding author
Corresponding author: Liyun Zhao (zhaoliyun@bnu.edu.cn);  John C. Moore
(john.moore.bnu@gmail.com)
**Abstract**
Basal thermal conditions play an important role in ice sheet dynamics, and they are
sensitive to geothermal heat flux (GHF). Here we estimate the basal thermal conditions,
including basal temperature, basal melt rate, and friction heat underneath the Lambert-
Amery glacier system in east Antarctica, using a combination of a forward model and
an inversion from a 3D ice flow model. We assess the sensitivity and uncertainty of
basal thermal conditions using six different GHFs. We evaluate the modelled results
using all available observed subglacial lakes. There are very large differences in
modelled spatial pattern of temperate basal conditions using the different GHFs. The
two most-recent GHF fields inverted from aerial geomagnetic observations have higher
values of GHF in the region, produce a larger warm-based area, and match the observed
subglacial lakes better than the other GHFs. The fast flowing glacier region has a lower
modelled basal friction coefficient, faster basal velocity, with higher basal frictional
heating in the range of 50-2000 mW m$^{-2}$ than the base under slower flowing glaciated
areas. The modelled basal melt rate reaches ten to hundreds of mm per year locally in
Lambert, Lepekhin and Kronshtadtskiy glaciers feeding the Amery ice shelf, and ranges
from 0-5 mm yr$^{-1}$ on the temperate base of the vast inland region.

**1 Introduction**
The Lambert-Amery system in East Antarctica is believed to be relatively stable against
climate change and has changed little over several decades of observations (King et al.,
2007). However, there is also evidence of extensive subglacial rifts and lakes. Jamieson
et al. (2016) report a large subglacial drainage network, suggesting that the region could
respond rapidly to changes in basal water supply or, potentially to surface forcing.

Extensive ice penetrating radar has been collected recently over Princess Elizabeth





Land (PEL; Fig. 1d), including the eastern part of the Lambert-Amery system (Cui et
al., 2020a). This fills in large data gaps from older surveys, and provides the basis for
our study. The radar surveys reveal ~1100 km long canyons (Fig. 1) that are incised
hundreds of meters deep into the subglacial bed that extend from the Gamburtsev
Subglacial Mountains (GSM) to the coast of the Western Ice Shelf (WIS). Li et al. (2021)
used airborne magnetic survey collected alongside the radar, which when combined
with the radar ice thicknesses and estimated depths at which the bedrock reaches its
Currie temperature, can be inverted for geothermal flux. This higher resolution data set
(Li et al., 2021) infers a larger heat flux than previous estimates in this region.
Furthermore recently discovered subglacial lakes, including potentially the second
largest subglacial lake in Antarctica, adds evidence for more widespread basal melting
in the region than was thought based on the much sparser earlier survey data (Cui et al.,
2020b). The complex subglacial topography, relatively high geothermal heat flux and
subglacial lakes imply a complex distribution of basal thermal condition and subglacial
water network. This heterogenous basal condition will have shaped much of ice flow
and the mass balance of the Lambert-Amery system. This motivates us to investigate
how the basal thermal condition inferred from the new high-resolution topography
dataset reconciles with surface ice velocities and existing geothermal heat flow maps.
Ice temperature is an important factor in the rheology of ice (Budd et al., 2013) and ice
flow. Whether the basal ice is at the melting point influences the movement of the ice
flow to a great extent. Ice at the melting point can lead to water lubricating the ice/bed
interface or saturating any sediment till layer and facilitating higher ice velocities via
basal sliding. This forms the basis for making inferences on basal conditions via surface
observations (Pattyn, 2010), or relict landforms (e.g. Näslund et al., 2005). Any
meltwater will tend to flow along hydraulic gradients, and accumulate in local
depressions (Fricker et al., 2016). The ice temperature is controlled by the
deformational heat generated from strain within the ice, the lateral advection of heat
due to ice motion and the descent rate of ice from the surface, the conduction of heat
through the ice and frictional heating from basal sliding.
Ice sheet models are useful tools to simulate the dynamic evolution of the ice sheet and
estimate its mass balance. Ice temperature is hard to evaluate because of the scarcity of
in-situ measurements, typically obtained from boreholes that are very rarely drilled
through the Antarctic ice sheet. GHF is an important boundary condition for ice
temperature, and is generally the largest source of uncertainty. Hence geophysical
survey methods are used to indirectly map GHF. To date GHF datasets have been
estimated from seismic models (Shapiro and Ritzwoller, 2004; An et al., 2015; Shen et
al., 2020), derived from airborne magnetic surveys (Li et al., 2021; Martos et al., 2017)
and satellite geomagnetic data (Maule et al., 2005; Purucker, 2013).
Large scale studies on the dependence on GHF of the Greenland (Rezvanbehbahani et
al., 2019) and Antarctica ice sheet (Pattyn, 2010) have inferred ice and basal
temperatures. Regionally, the rapidly retreating Thwaites and Pope glaciers in the
Amundsen Sea sector of West Antarctica is being facilitated by the high heat flow in
the underlying lithosphere (Dziadek et al., 2021). In the Lambert-Amery glacial system,
Pittard et al. (2016) suggest that ice flow is most sensitive to the spatial variation in the
underlying GHF near the ice divides and along the edges of the ice streams.
In this study, we simulate ice basal temperatures and basal melt rates in the Lambert-
Amery system using the new high-resolution digital elevation model, along with six
different published GHF maps as forcing for an off-line coupling between a basal
energy and water flow model and a 3D full-Stokes ice flow model. We evaluate the
quality of the resulting basal temperature field incorporating the Stokes model estimates
of ice advection, strain and frictional heating under the different GHF maps using all
available observed subglacial lakes and surface velocities. Hence, we make inferences
on which GHF maps yield the best match with observations in the region.
**2 Regional Domain and Datasets**
Our modeled domain is in the Lambert-Amery system. It consists of two drainage
basins: the Lambert Glacier Basin, the American Highland Basin, along with half of
Amery Ice Shelf (Fig. 1). The 2D domain boundary outlines are defined by the inland
ice catchment basin boundary, the central streamline, and the ice front of Amery Ice
Shelf. The inland sub-basin and the central streamline of the Amery Ice Shelf were
chosen as boundaries because the mass flux across them is assumed to be zero by
definition.

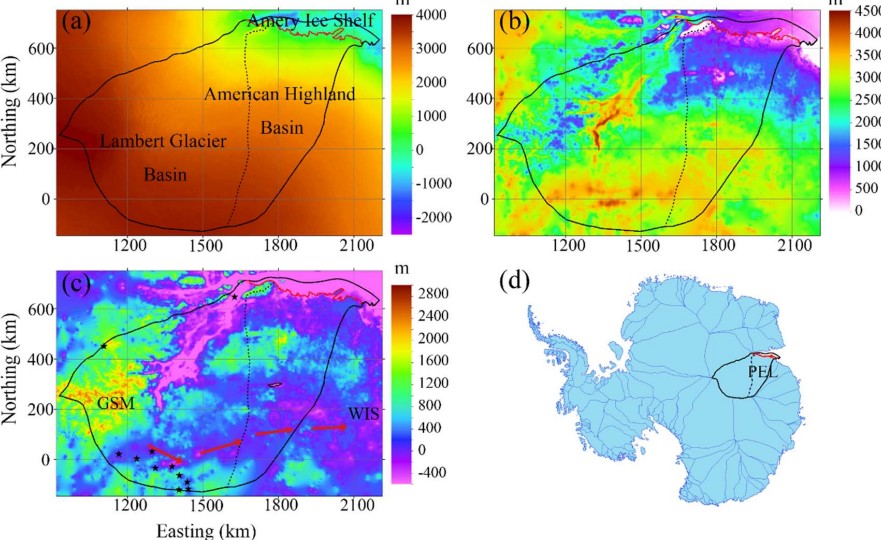

Fig. 1. The domain topography and location with domain boundary overlain. (a) surface elevation;
(b) ice thickness; (c) bed elevation; (d) the location of our domain in Antarctica. The solid black
curve is the outline of the study domain, including the central streamline of Amery ice shelf and the
boundary of inland sub-basins based on drainage-basin boundaries defined from satellite ice sheet





surface elevation and velocities (Mouginot et al., 2017; Rignot et al., 2019). The red curve is part of
the grounding line of Amery ice shelf. The dotted black curve is the dividing line between Lambert
Glacier Basin and the American Highland Basin.  The black stars in (c) denote the locations of
observed subglacial lakes, and the area surrounded by the black line in American Highland Basin in
(c) is the potentially second largest subglacial lake in Antarctic. The red arrows in (c) indicate the
routing through the deep subglacial canyon system from GSM to WIS.
The surface elevation, bedrock elevation, and ice thickness are from MEaSUREs
BedMachine Antarctica, version 2 with a resolution of 500 m (Morlighem et al., 2020).
Additional ice thickness data from Cui et al. (2020a) were added to further constrain
the bed topography beneath the grounded ice (Table 1). The bed elevation is calculated
using upper surface elevation minus ice thickness.
The surface ice velocity data are obtained from MEaSUREs InSAR-based Antarctic ice
velocity Map, version 2 with resolution of 450 m (Rignot et al., 2017). Data were largely
acquired during the International Polar Years 2007 to 2009, and between 2013 and 2016.
Additional data acquired between 1996 and 2016 were used as needed to maximize
coverage.
Ice sheet surface temperature data are prescribed by ALBMAP v1 with resolution of 5
km (Le Brocq et al., 2010) and come from monthly estimates inferred from AVHRR
data averaged over 1982-2004. Subglacial lake locations are from the fourth inventory
of Antarctic subglacial lakes(Wright and Siegert, 2012), with the addition of the newly
discovered lakes (Cui et al., 2020b).
Six GHF datasets (Fig. 2; Table 2) are used in this study. All the datasets are interpolated
into the same 2.5 km resolution.
Table 1 Datasets used in this study.

| Variable name | Dataset | Resolution | Reference |
|---|---|---|---|
| surface elevation, bedrock elevation, and ice thickness | MEaSUREs BedMachine Antarctica version 2 | 500 m | Morlighem et al., 2020; Cui et al., 2020 |
| surface ice velocity | MEaSUREs InSAR-based Antarctic ice velocity Map, version 2 | 450 m | Rignot et al., 2017 |
| surface temperature | ALBMAP v1 | 5 km | Le Brocq et al., 2010; |
| subglacial lakes location | The fourth inventory of Antarctic subglacial lakes | ----- | Wright and Siegert, 2012; Cui et al., 2021 |


Table 2 The six GHF datasets used in this study.

| GHF map | Reference | Method | Mean (mW m$^{-2}$) | Range (mW m$^{-2}$) |
|---|---|---|---|---|
| Martos | Martos et al., 2017 | airborne geomagnetic data | 72 | 47-90 |
| Shen | Shen et al., 2020 | seismic model | 50 | 43-59 |
| An | An et al., 2015 | seismic model | 55 | 40-66 |
| Shapiro | Shapiro and Ritzwoller, 2004 | seismic model | 54 | 45-58 |
| Purucker | Purucker, 2013 | Satellite geomagnetic data | 47 | 26-47 |
| Li | Li et al., 2021 | airborne geomagnetic data | 72 | 52-90 |

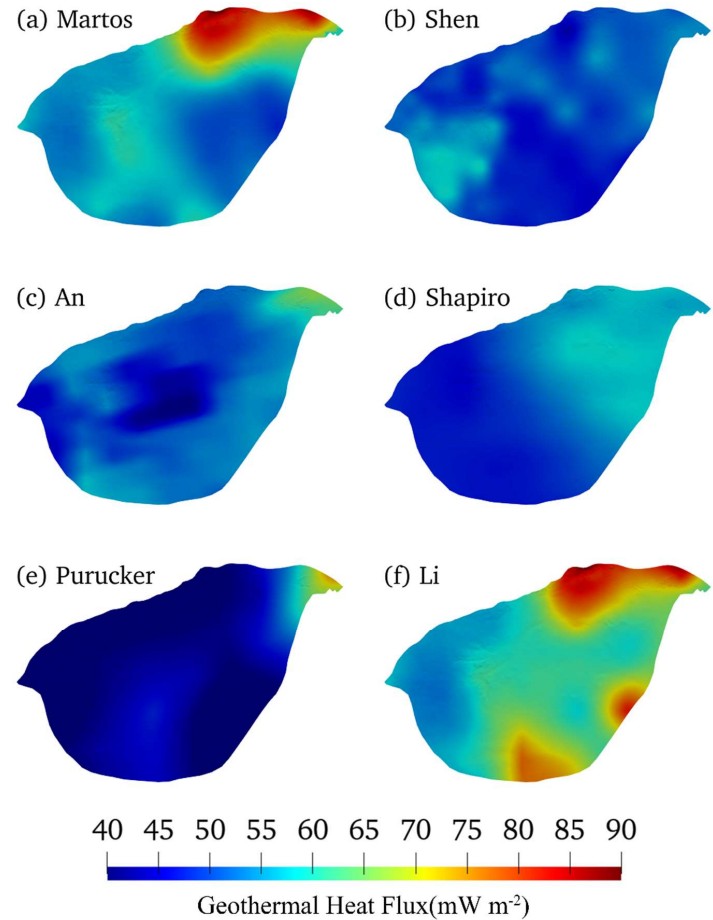

Geothermal Heat Flux(mW m$^{-2}$)


Fig. 2. The spatial distribution of GHF over our domain as described in Fig. 1. See Table 2 for the
GHF map details.


**3 Model**

Our goal is to infer the basal thermal condition, including basal temperature and basal
melt rate in the domain. Geothermal heat flux, internal heat conduction and basal
friction heat are the main heat sources that determines the basal thermal condition.
Therefore, we need to model both ice flow velocity and stress for basal friction heat and
ice temperature for internal heat conduction.

We use an inverse method implemented in a full-Stokes model, Elmer/Ice, to estimate
ice flow velocity and stress, infer the basal friction coefficient and obtain the basal
friction heat. A proper initial ice temperature is needed in the inverse method. To get it,
we use a forward model that consists of an improved Shallow Ice Approximation (SIA)



thermomechanical model with a subglacial hydrology model (Wolovick et al., 2021).
The forward model uses the velocity direction and basal slip ratio from the full-Stokes
inverse model to constrain its solution. We do steady state simulations by coupling the
two models.

### 3.1 Forward Model

The forward model consists of a thermomechanical steady state model using an
improved Shallow Ice Approximation (SIA) in equilibrium with the subglacial
hydrological system (Wolovick et al., 2021), described in Sections 3.1.1-3.1.3. It has
internal consistency between three components: ice flow, ice temperature, and basal
water flux. The numerical model requires three coupled components to be consistent
with one another: (1) Integration for balance flux and englacial temperature downhill
in the ice surface, (2) Integration for basal water flux and freezing rate downhill in the
hydraulic potential, and (3) Rheology and shape function computations to determine
the distribution of ice flux and shear heating. These three components are solved within
a large fixed-point iteration. In our coupling scheme, components (1) and (3) are
constrained by the velocity direction and basal sliding ratio computed by the full-Stokes
inverse model. The simulation is done on a finite difference mesh with resolution of 2.5
km.
### 3.1.1 Balance Flux and Thermal Model
The mass balance of the ice sheet is given by,
$$\nabla \cdot (\overline{u}H) = a - m, \tag{1}$$
where $\overline{u}$ is the vertically averaged horizontal velocity, H is the ice thickness, $a$ is
surface accumulation rate and $m$ is basal melt rate, both expressed as ice equivalent
thickness per unit time. In most of the domain, the direction of the horizontal velocity
vector is taken from the full-Stokes Elmer/Ice model, but the magnitude of horizontal
velocity is allowed to vary to ensure exact mass conservation for a given surface
accumulation rate and basal melt forcing. Near the domain edges the velocity direction
in Elmer/Ice is unreliable, and the smoothed surface gradient is used to provide velocity
direction at those locations instead. The magnitude of horizontal velocity is determined
using a balance flux algorithm (e.g., Budd and Warner, 1996). The integration order is
taken from the smoothed ice surface elevation, with local corrections to ensure that no
grid cells depend on values downstream of themselves. Once the column-average
horizontal velocity in a given grid cell is known, the vertical distribution of horizontal
velocity in the ice column is calculated by:
$$\boldsymbol{u}(x, y, z) = \overline{u}(x, y)\hat{u}(x, y, z), \tag{2}$$
where $u = (u_x, u_y)$ is the horizontal velocity vector and $\hat{u}$ is a dimensionless scalar
shape function for horizontal velocity (section 3.1.3). The shape function is taken from
the last iteration and interpolated to the edge of the mesh. Once the vertical distribution
of horizontal velocity is known, the vertical velocity is calculated from the
incompressibility condition by,



$$w(z) = -m - \int_0^z \nabla \cdot u \, dz', \tag{3}$$

where $w$ is the vertical component of velocity.

After obtaining all the three components of the velocity, the ice column temperature can
be calculated from the conservation of energy,
$$-\frac{d}{dz}\left(k(T)\frac{dT}{dz}\right) + \rho_i \vec{u} \cdot \nabla\left(c_{p,i(T)}T\right) = 4\eta\dot{\varepsilon}_E^2, \tag{4}$$

where $T$ is temperature, $k(T)$ is the temperature-dependent thermal conductivity of ice,
$\rho_i$ is the density of ice, $c_{p,i(T)}$ is the temperature-dependent specific heat capacity of ice,
$\vec{u}$ is the full (3 component) velocity vector, $\eta$ is the ice viscosity and $\dot{\varepsilon}_E$ is the effective
strain rate.

For the thermal boundary condition, at the surface, we use the Dirichlet condition where
the temperature is equal to the surface temperature. At the bottom of ice shelves, we set
basal temperature as pressure melting point. At the bed of grounded ice, the boundary
condition can be either Dirichlet or Neumann condition depending on the basal melting
and subglacial water conditions. The logical conditions are given by,
$$k(T)\frac{dT}{dz} = G, \quad \text{for } T < T_m \text{ and } m = 0; \tag{5}$$

$$T = T_m, \quad \text{for } m \neq 0, \tag{6}$$

Where $T_m$ is the pressure-dependent melting temperature, $G$ is GHF, taking six GHF
datasets listed in Table 2. The thermal condition will switch from Neumann (Eq 5) to
Dirichlet (Eq 6) if the basal temperature exceeds the pressure-dependent melting point.
The opposite switch from Dirichlet to Neumann is determined by the hydrology model,
if there is insufficient water input to supply a large freezing rate. The basal melt rate
here can be either positive or negative representing melting or freezing; when it is
negative, the freezing must be balanced by an influx of water.

The thermally determined melt rate is,
$$\rho_i L m = G - k(T)\frac{dT}{dz} + \vec{u}_b \tau_b - \overrightarrow{q_w} \cdot \left(\nabla\phi + \rho_w c_{p,w}\beta\nabla P\right), \tag{7}$$

where $L$ is the latent heat of fusion, $\overrightarrow{q_w}$ is the flux of water along the basal plane, $\phi =$
$\rho_i gH + \rho_w gB$ is the hydraulic potential, $\rho_w$ is the density of water, $c_{p,w}$ is the specific
heat of water, $\beta$ is the pressure coefficient of the melting point, and $P = \rho_i gH$ is the
overburden pressure of the ice sheet. The final term in Eq 7 represents the combined
effect of viscous dissipation and sensible heat changes within the water system and can
potentially give rise to glaciohydraulic supercooling.

**3.1.2 Basal Hydrology Model**
The water flux is determined from mass conservation,
$$\nabla \cdot \overrightarrow{q_w} = \frac{\rho_i}{\rho_w} m, \tag{8}$$

where we have included the density ratio to convert melt rate from ice-equivalent



thickness to water-equivalent thickness. The water flux is computed in a similar style
of balance-flux calculation as the ice flux, where the flux vector is assumed to point
downhill in hydraulic potential and Eq 8 is integrated downhill to determine flux
magnitude. The water flow is governed by hydraulic potential $\phi$. We fill closed basins
in the hydraulic potential before running the model to ensure that we have a continuous
flow path all the way to the margins of the domain. As long as water flux magnitude
remains positive (that is, directed down-potential), the hydrology model uses the
thermally determined melt/freeze rate from Eq 7. In the event that the balance flux
calculation would yield negative water flux (that is, water flow directed up-potential),
the hydrology model switches the grid cell from Dirichlet back to Neumann, and the
limiting freezing rate is determined by rearranging Eq 8 to solve for the value of $m$ that
results in zero flux leaving the grid cell. In grid cells that receive no water input from
upstream, this merely means that the melt/freeze rate is set to zero and the basal
boundary condition can be given by Eq 5 without complication, but for grid cells at the
termination of a water network, a special partially frozen condition must be used.
When a water network terminates by freeze-on, we have grid cells in which the freezing
front penetrates partially through the grid cell but not completely. To respect both mass
and energy conservation, it is necessary for there to be a nonzero freezing rate and
nonzero water flux entering these grid cells despite the fact that their average
temperature is below the melting point. For these partially frozen cells, the freezing rate
is determined by the water supply through Eq 8 as described above. That freezing rate
is associated with a release of latent heat, which must be accounted for by the thermal
model. The hydrology model, therefore sets these grid cells to a Neumann condition,
but instead of being taken from Eq 5, the basal temperature gradient is determined by
rearranging Eq 7 to solve for $\frac{dT}{dz}$. The basal temperature in these grid cells is thus not
fixed to the melting point, but it nonetheless is higher than it otherwise would be
because of the release of latent heat at the termination of the water network.
**3.1.3 Rheology and Shape Function Model**
The shape function determines the distribution of horizontal velocity with depth. The
effective viscosity of the ice is given by,

$$\eta = \frac{1}{2}\big(A(T)\big)^{-\frac{1}{n}}\dot{\varepsilon}_E^{\frac{1-n}{n}}, \tag{9}$$

where $A(T)$ is the temperature-dependent rate factor calculated using an Arrhenius
equation (Cuffey and Paterson, 2010),

$$A(T) = A_0 \exp\left(\frac{-Q}{RT}\right), \tag{10}$$

where $A_0$ is the prefactor, $Q$ is the activation energy, $R$ is the universal gas constant.
$n = 3$ is the rheological exponent for ice. The effective strain rate $\dot{\varepsilon}_E$ is given by,

$$\dot{\varepsilon}_E = \sqrt{\dot{\varepsilon}_{xy}^2 + \dot{\varepsilon}_{xz}^2 + \dot{\varepsilon}_{yz}^2 + \frac{1}{2}\big(\dot{\varepsilon}_{xx}^2 + \dot{\varepsilon}_{yy}^2 + \dot{\varepsilon}_{zz}^2\big)}. \tag{11}$$






Once we have the viscosity, we can compute the shape function. We use the results of
the full-Stokes inverse model to constrain the slip ratio, $\hat{u}_b = u_b/\bar{u}$. We then assume
the shear stress between the bed and the surface varies linearly, and then use the
relationship between stress and strain rate to get the vertical gradient of horizontal
velocity, $\frac{du}{dz} = \sigma_b \frac{1-\hat{z}}{\eta}$. Integrating this expression up from the bed, normalizing to unit
amplitude, and canceling the common factor, we get the shape
function,
$$\hat{u}(\hat{z}) = \hat{u}_b + (1 - \hat{u}_b)\frac{\int_0^{\hat{z}}\frac{1-\hat{z}'}{\eta(\hat{z}')}d\hat{z}'}{\int_0^1\frac{1-\hat{z}'}{\eta(\hat{z}')}d\hat{z}'},$$
(12)

where $\hat{u} = u/\bar{u}$ is the shape function for horizontal velocity, $\hat{z} = (z - B)/H$ is
normalized elevation.

**3.2 Inverse Method with full-Stokes Model**
The spatial distribution of basal friction in the domain is modelled by an inverse method
using three-dimensional the full-Stokes model, Elmer/Ice, an open source finite element
method package(Gagliardini et al., 2013). The inverse method is based on adjusting the
spatial distribution of the basal friction coefficient to minimize the misfit between
simulated and observed surface velocities. The modelled velocity is obtained by solving
the full-Stokes equation, which includes conservation equations for both the momentum
and mass of the ice,
$$div\,\boldsymbol{\tau} - grad p = \rho_i g,$$ (13)
$$div\,\vec{v} = 0,$$ (14)
where $\tau$ is the deviatoric stress tensor, $p$ is the isotropic pressure, $\rho_i$ is ice density, $g$ is
the acceleration due to gravity (0, 0, -9.81) m·s$^{-2}$, $\vec{v} = (u, v, w)$ is ice velocity.
According to Glen's flow relation, deviatoric stress is related to the deviatoric part of
the strain rate tensor, $\dot{\varepsilon}_E$, which can be described by $\tau = 2\eta\dot{\varepsilon}_E$, where $\eta$ denotes ice
viscosity, given by Eq 9, is sensitive to the flow rate factor $A(T)$ given by Eq 10. Ice
temperature distribution is from the modelled result of forward model in section 3.1.

**3.2.1 Mesh Generation and Refinement**
Firstly, we use GMSH (Geuzaine and Remacle, 2009) to generate an initial 2-D
horizontal footprint mesh with the boundary described in section 2. Then we refine the
mesh by an anisotropic mesh adaptation code called the Mmg library
(http://www.mmgtools.org/). The resulting mesh is shown in Fig. 3 and has minimum
and maximum element sizes of approximately 1000 m and 8000 m. The 2-D mesh is
then vertically extruded using 10 equally spaced, terrain following layers.

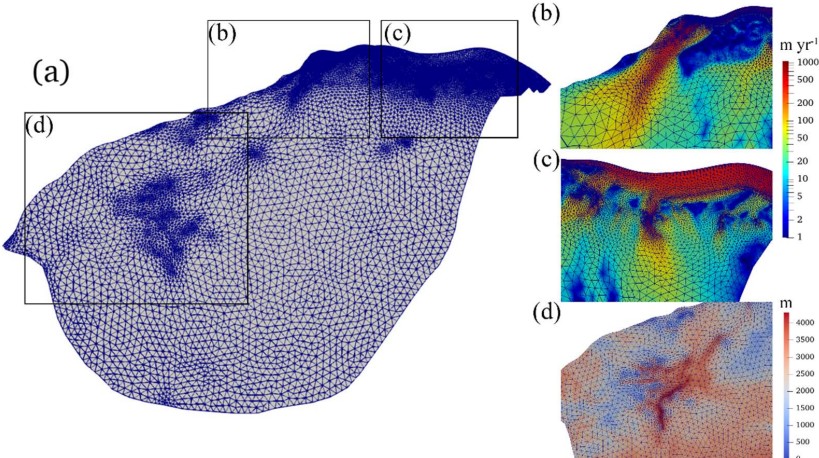

Fig. 3. The refined 2-D horizontal domain footprint mesh (a). Boxes outlined in (a) are shown in detail overlain with surface ice velocity in (b) and (c), and with ice thickness in (d).

### 3.2.2 Boundary Condition

The ice surface is assumed to be stress-free. At the ice front, the normal stress under the sea surface is equal to the hydrostatic water pressure. On the lateral boundary, the normal stress is equal to the ice pressure applied by neighboring glaciers and the normal velocity is assumed to be 0. The bed for grounded ice is assumed to be rigid, impenetrable, and fixed over time.

The normal basal velocity is set to 0 at the ice-bed interface. The Weertman sliding law is used to describes the relationship between the basal sliding velocity, $u_b$, and the basal shear force, $\tau_b$, on the bottom of grounded ice,

$$\tau_b = C\ \vec{u}_b. \tag{15}$$

To avoid non-physical negative values, $C = 10^\beta$ is used in the simulation. We call β the basal friction coefficient rather than $C$. $C$ is initialized to a constant value of $10^{-4}$ MPa m$^{-1}$ yr (Gillet-Chaulet et al., 2012), and then replaced with the inverted $C$ in subsequent inversion steps.

### 3.2.3 Surface Relaxation

We relax the free surface of the domain by a short transient run to reduce the non-physical spikes in initial surface geometry (Zhao et al., 2018). The transient simulation period here is 0.5 yr with a timestep of 0.01 yr.

### 3.2.4 Inversion for Basal Friction Coefficient

Taking the results from surface relaxation as initial condition, we use an inverse method to retrieve the basal friction coefficient, the deviatoric stress field and ice velocity field. The inverse method is to adjust the basal friction coefficient $C$ to minimize the value of the cost function (Morlighem et al., 2010), which is defined as the difference between the simulated surface velocity and the observed,




$$J_0 = \int_{\Gamma_s} \frac{1}{2}(|u| - |u^{obs}|)^2 d\Gamma, \tag{16}$$

where $\Gamma_s$ is the ice surface, $u$ and $u^{obs}$ are the simulated and observed surface velocities.

To avoid over-fitting of the inversion solution to non-physical noise in the observations,
a regularization term,
$$J_{reg} = \frac{1}{2}\int_{\Gamma_s}\left(\left(\frac{\partial c}{\partial x}\right)^2 + \left(\frac{\partial c}{\partial y}\right)^2\right)d\Gamma, \tag{17}$$

is added to the cost function, then the total cost function is defined as,
$$J_{tot} = J_0 + \lambda J_{reg}, \tag{18}$$
where $\lambda$ is a positive regularization weighting parameter. An L-curve analysis (Hansen
and Johnston, 2000) has been done for inversions to find the optimal $\lambda$ by plotting the
term $J_{reg}$ as the function of $J_0$. The optimal value of $10^{10}$ is chosen for $\lambda$ to minimize $J_0$.

**3.2.5 Basal Melt Rate**
Based on the inverted basal velocity and basal shear force, we can calculate the basal
friction heat. Then we can produce the basal melt rate using the thermal equilibrium as
follows:
$$M = \frac{G + \vec{u}_b \tau_b - k(T)\frac{dT}{dz}}{\rho_i L}, \tag{19}$$

where M is the basal melt rate. The ice-bed interface gets heat through GHF and friction
heat but loses heat from upward heat conduction.

**4 Simulations and Results**
**4.1 Experimental Design**
We design the coupled simulations by an 8-step scheme by coupling the forward model
and inverse model:
1. We run the forward model with the velocity direction taken from a mixture of the
surface gradient and surface velocity observations, and get an initial modelled
englacial temperature.
2. We do surface relaxation in Elmer/Ice with the englacial temperature from step
1.
3. Taking the results from step 2 as the initial state, we do inversion simulation by
Elmer/Ice using the modeled englacial temperature from step 1, to get a modelled
surface velocity best fit to the observed surface velocity. The modelled surface
velocity will remove some artifacts in the observed field.
4. We run the forward model using the velocity directions from Elmer-Ice, and get
an updated modelled englacial temperature.
5. We run the inverse method in Elmer/Ice with the improved englacial temperature
from step 4, and get an updated modelled velocity.





6.   We run the forward model again using the ratio of basal sliding to column-
average velocity in Elmer/Ice from step 5 to constrain the slip ratio, and get a
further updated basal temperature.
7.   We run the inverse method again in Elmer/Ice with the improved englacial
temperature from step 6, and get an updated modelled velocity and stress.
8.   We analyze the modelled results in step 7, calculate basal friction heat and basal
melt rate.
We use six sets of GHF in basal thermal condition in the forward model, and obtain six
sets of englacial temperature used in the inverse model. Correspondingly, we call the
six experiments: Martos, Shen, An, Sr, Purucker and Li.

**4.2 Improvement of Basal Friction Coefficient**

Basal friction in reality depends on basal temperature, i.e., it is relatively large on cold
beds since the ice is frozen, and small on warm bed where basal temperature reaches
pressure-melting point allowing the ice to slide. However, in the inverse model, basal
friction coefficient (Eq 15) is adjusted to match velocity observations without regard to
basal temperature,  which leads to unrealistic noise manifested as local spikes in
modelled basal friction heat .

We improve the parameterization of $\beta$ via $C$ in Eq 15 (Section 3.2.2) by considering
basal temperature $T_{bed}$,

$$\beta_{new} = \beta_{old} + \alpha(T_M - T_{bed}), \tag{20}$$

where $\beta_{old}$ is modelled by inverse model, $\alpha$ is a positive factor to be tuned, $T_M$ is
pressure-dependent melting temperature. $\beta_{new}$ equals $\beta_{old}$ at a warm bed with
temperate ice, and is larger than $\beta_{old}$ at a cold bed with ice temperature lower than $T_M$.
We tune $\alpha$ in the range of [0.1, 2] with an interval of 0.1, and find the local spikes in
modelled friction heat become less as $\alpha$ increases from 0.1 to 1, and keep almost the
same with $\alpha$ from 1 to 2. Therefore, we take $\alpha$ to be 1, and use the parameterization of
$\beta_{new}$ in Eq 20 in all the simulations. Using Eq 20 does not change the modelled surface
velocity in the interior region.

**4.3 Simulation Results**

**4.3.1 Ice Velocity**
In the inverse method, the modeled the surface velocity matches best to the observed
surface velocity. Therefore, we get very similar distributions of modeled velocity field
using different GHFs. Fig. 4 shows the modelled velocity in the Martos experiment as
an example. The modeled surface velocity shows spatial similarities to the observed
surface velocity (Fig. 4a, b). Three fast-flowing outlet glaciers (Lambert Glacier,
Lepekhin Glacier and Kronshtadtskiy Glacier) deliver ice to the ice shelf. The velocity
of the Lambert glacier exceeds 800 m yr$^{-1}$ at the grounding line. The Lepekhin Glacier



and the Kronshtadtskiy Glacier have maximum flow velocities of about 200 and 400 m
yr$^{-1}$ at their grounding lines, respectively. Regions with large differences between
modeled and observed surface velocity occupy a small fraction of the whole area (Fig.
4c) and are associated with high velocity gradients. Ice velocity decreases with depth.
Fig. 4d shows modeled basal ice velocity. The maximum basal velocity on Lambert
Glacier exceeds 500 m yr$^{-1}$, and maximum basal velocities on Lepekhin Glacier and the
Kronshtadtskiy Glacier reach about 150 and 200 m yr$^{-1}$ at the grounding line.

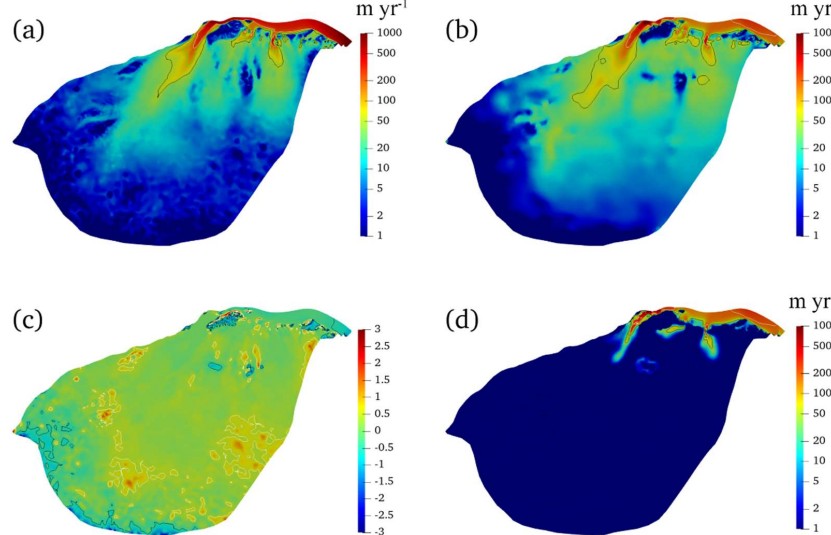

Fig. 4. (a) Observed surface velocity, (b) modeled surface velocity in the Martos experiment, (c)
difference of modeled and observed surface velocity plotted as log$_{10}$(modeled/observed), (d)
modeled basal velocity. The black, cyan and white solid lines in (a), (b), and (d) represent speed
contours of 50, 100 and 200 m yr$^{-1}$, respectively. The white lines in (c) represent contours of 0.5,
and the black lines represent contours of -0.5. The three fast-flowing outlet glaciers in plot (a) from
left to right are Lambert, Lepekhin and Kronshtadtskiy glaciers.

### 4.3.2 Basal Ice Temperature and Heat Conduction

In Fig. 5 we show the modelled basal temperature from the six experiments. There are
significant differences in the modelled distribution of warm base (basal temperature
reaching the pressure melting point) using different GHFs. The basal temperature is
highly dependent on the GHF. In the Li experiment which has high GHF over the
domain, the basal temperatures over most of the domain reaches the melting point, and
the area of warm base is the largest. The Martos experiment with the second high GHF
yields the second largest area of warm base and the Purucker experiment with the
coldest GHF gives the smallest area which is concentrated around the fast-flowing ice.
All experiments display cold basal temperatures at the southwest of Lambert Glacier
Basin, where there are subglacial mountains.



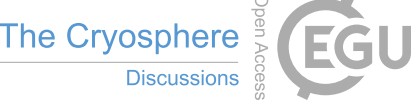

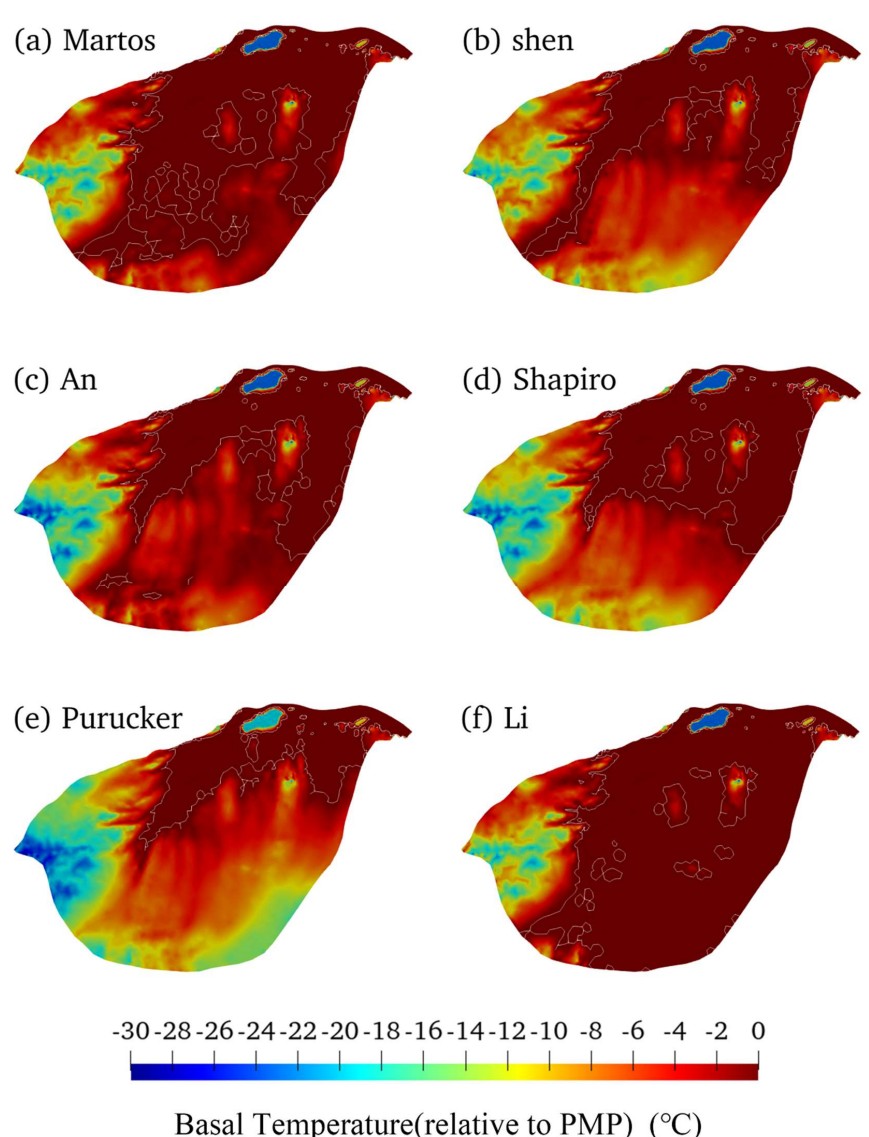


Fig. 5. Modelled basal temperature relative to pressure melting point, (a) to (f) corresponding to the
GHF (a) to (f) in Fig. 2. The ice bottom at the pressure-melting point is delineated by a white contour.



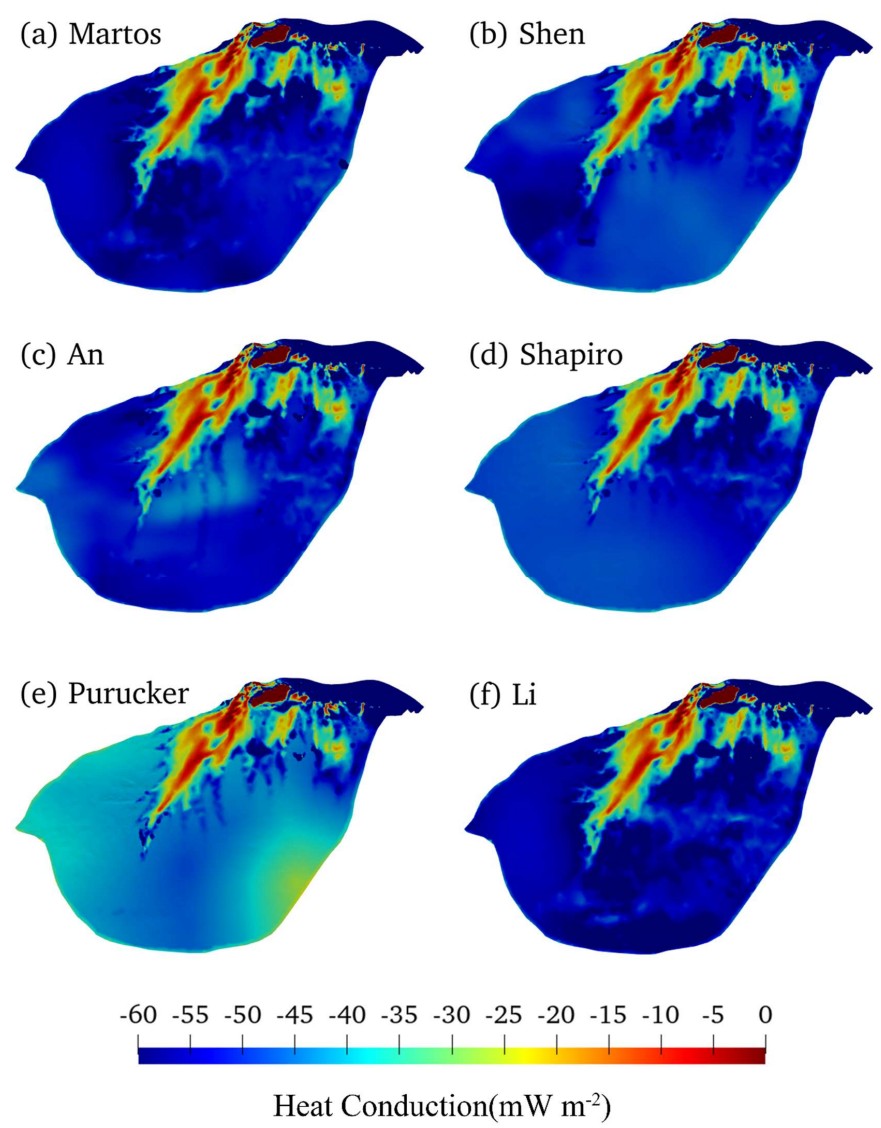


Fig. 6. Modelled basal heat conduction (unit: mW m$^{-2}$). (a) to (f) corresponding to the GHF (a) to
(f) in Fig. 2.

Fig. 6 show the modelled heat conduction in the six experiments. In the fast-flowing
tributaries (Fig. 4a), the upward heat conduction is lower than 0-30 mW m$^{-2}$ in all
experiments. For vast inland areas, most experiments yield high upward heat
conduction in the range of 45-60 mW m$^{-2}$ except for the Purucker experiment which
has lower values around 30-45 mW m$^{-2}$.





### 4.3.3 Basal Friction Heat


Fig. 7 shows the modelled friction heating maps in six experiments. As expected, basal
friction heat is high in fast-flowing regions. The fast-flowing tributaries have friction
heating over than 50 mW m$^{-2}$ and reach 2000 mW m$^{-2}$ at the grounding line.

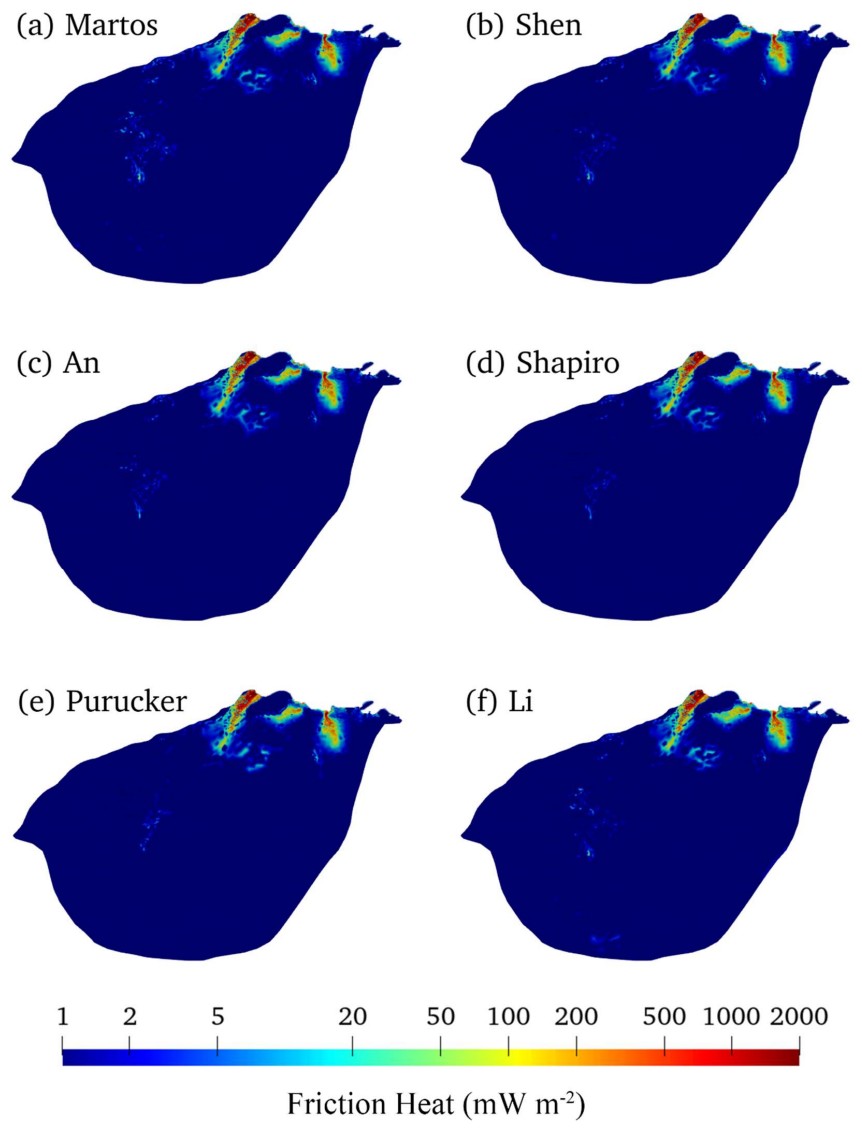


Fig. 7. Modeled basal friction heat (unit: mW m$^{-2}$), (a) to (f) corresponding to the GHF (a) to (f) in
Fig. 2, respectively.





### 4.3.4 Basal Melt Rate

We get the basal melt rate using the thermal balance equation (Eq 19). Fig. 8 shows the modelled basal melt rate in the six experiments using different GHF. Regions with basal melt rate coincide with a warm base where basal temperatures reach pressure-melting point. There are significant differences in area with basal melting among the six experiments due to large variability in GHF. The Li and Martos experiments yield the largest area with basal melting. In contrast, Purucker experiment gives the least area with basal melting (Fig. 8).

The modelled basal melt rate is below 5 mm yr$^{-1}$ in the parts of the vast inland region that are warm based. Higher basal melt rate occurs in fast-flowing regions (Fig. 8Fig. 8) where frictional heat is high (Fig. 7), despite the differences in GHF (Fig. 4). Maximal basal melting rate is above 10 mm yr$^{-1}$ near the grounding line, reaching 500 mm yr$^{-1}$ at the grounding line of the central flowline running onto Amery ice shelf. Thus, in fast-flowing regions, frictional heat is the dominant factor rather than GHF, consistent with Larour et al., (2012) who noted that slower flowing ice in the interior of the ice sheet will be more sensitive to the GHF, but frictional heat dominates GHF in regions of fast ice flow.

We use the positions of observed subglacial lakes to validate simulated regions with basal melting (Fig. 8). The Li experiment gives the best fit between the observed subglacial lakes and the  modelled warm base region (Fig. 8f). The modelled warm base covers all the observed subglacial lakes in the domain, including the recently discovered second-largest subglacial lake in Antarctica (Cui et al., 2020b). The Martos experiment gives the next best fit (Fig. 8a), and the An experiment the third (Fig. 8c). The Shen experiment captures two subglacial lakes in the southwest of the domain (Fig. 8b), while the Shapiro experiment missed many known subglacial lakes in the southwest of the domain, but successfully captures the recently discovered second-largest subglacial lake (Fig. 8b, d). The Purucker experiment performs worst in recovering subglacial lake locations (Fig. 8e).

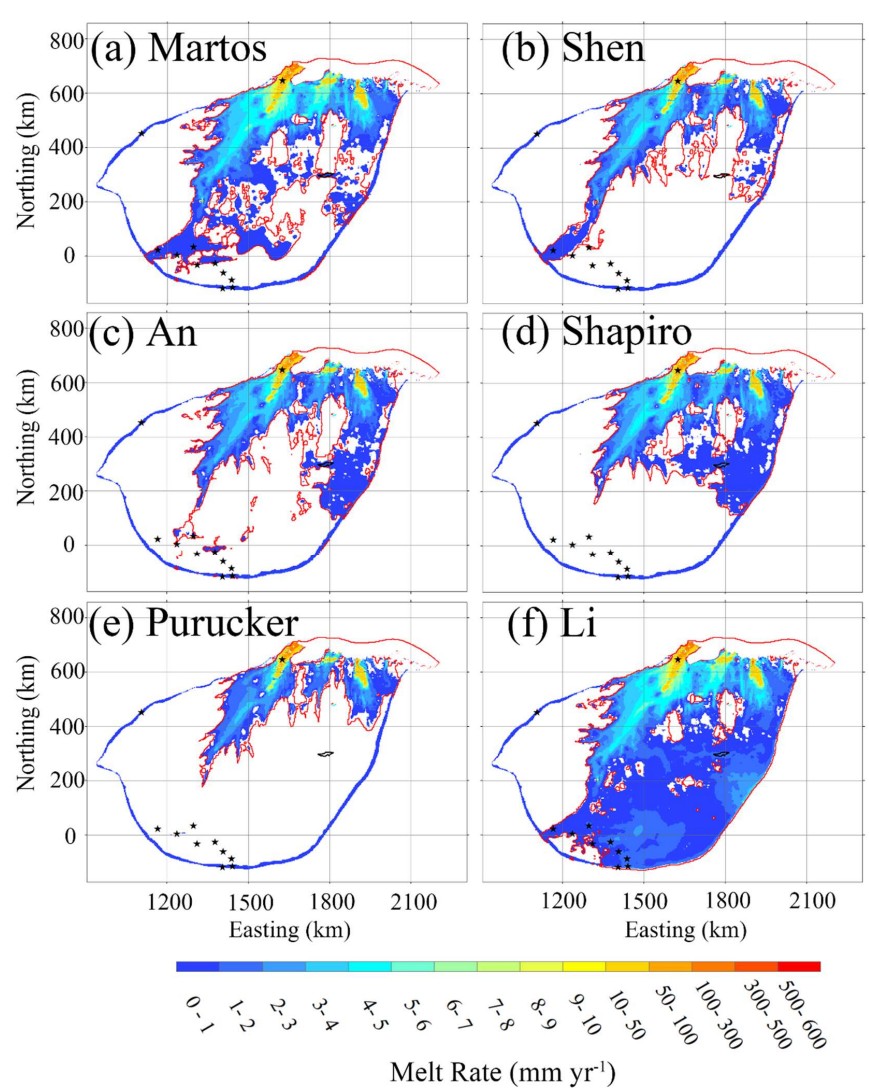


Fig. 8. Modelled basal melt rate (unit: mm yr⁻¹), (a) to (f) correspond to the GHF (a) to (f) in Fig. 2.
The ice bottom at pressure-melting point is surrounded by a red contour. The stars denote the
locations of observed subglacial lakes, and the area surrounded by the black line is the likely second
largest subglacial lake in Antarctica.

**5 Discussion**
GHF distribution largely governs basal thermal conditions.  Many previous studies on
basal temperature and basal melt have used the Shapiro, Fox, Purucker, and An datasets,
with few making use of the more recent Martos and Li fields. In this study, we find that
the Li and Martos experiments have higher GHF than the earlier datasets in the
Lambert-Amery domain and consequently have the largest area with warm base. The
warmer basal conditions best match the observed distribution of subglacial lakes.



However, it should be noted that observations of subglacial lakes are a one-sided
constraint: a model result that misses the observed lakes is clearly too cold, but if the
model puts warm-based conditions outside of the locations of the observed lakes, it is
not clear whether this is because the model is too warm, or if the subglacial water exists
in a form other than ponded lakes.
Our methodology builds on the earlier inversion method employed by Wolovick et al.
(2021). Specifically, we use the full-Stokes flow model in the inversion of basal friction
field rather than a simplified physics model. We also improve on the treatment of basal
friction field by imposing an increase in basal friction where the ice bed is colder than
the pressure melting point, and which increases with temperature difference from
freezing point. These modifications produce more physically meaningful results since
we expect frozen beds to have high basal friction. Hence, the basal friction field is
constrained by simulated temperatures in addition to producing the best fitting match
of simulated and observed surface velocities.
Van Liefferinge and Pattyn (2013) estimated basal temperature for the Antarctica ice
sheet using three GHF datasets (Fox, Sharpio, Purucker), and each of the datasets were
improved by the method in Pattyn (2010). Their modeled temperatures show spatial
similarities to the Purucker experiment field in our study. Pittard et al. (2016) did
sensitivity experiments of the Lambert-Amery glacial system based on 3 GHF fields
(Fox, An, Sharpio) using the ice dynamics model PISM, and found that modelled basal
temperature reached the pressure melting point only under the fast-flowing ice, with
maximum melting rates of 500 mm yr$^{-1}$ at places very close to the grounding line of the
central flowline onto the Amery ice shelf. This is similar to our modelled maximum
basal melt at similar locations in the six experiments. However, their modelled region
of basal melt is mainly confined to the Lambert glacier tributary and well-matches only
that of the Purucker experiment in our study.
We analyze the contribution of GHF and frictional heat to basal melt. The basal friction
is a significant heat sources only under fast-flowing ice. The GHF distribution in the
ice sheet connected to the ice shelf is much more homogeneous, but frictional heating
means that the melt rate in the fast-flowing ice is more than 10 times higher than that
in the slow-flowing ice. Thus slower flowing ice in the interior of the ice sheet is more
sensitive to the GHF, than fast-flowing ice (Larour et al., 2012).
GHF has its largest impact on the basal melt of the vast inland ice sheet. There are two
principle ways to constrain GHF: (1) direct measurement (2) inversion by multiple
geophysical methods. The GHFs used in this study are based on inversion of satellite
or aero magnetic data and seismic tomography. Direct observations of heat flux are
difficult to obtain in Antarctica, and satellite data are low resolution. The most efficient
methods is to invert the heat flux through aerial geomagnetic observation such as for
the Martos and Li GHF fields. However, there are still large data gaps in remote regions,
especially in PEL, leaving just inversion using satellite magnetic data with a lower





resolution. The Li et al. (2021), field uses the latest aeromagnetic data to estimate the
GHF in the PEL region and this gives higher values than derived previously.
To validate the modelled basal melt, we use the locations of detected subglacial lakes.
There may be many other undiscovered subglacial lakes beneath the study area, and
further discoveries would help us validate the model results, and possibly refine GHF
maps. In addition, further observational constraints with a two-sided sensitivity to ice
temperature, such as observations of subglacial freeze-on or measurements of englacial
attenuation, would help us to identify areas in which the GHF maps are too warm, in
addition to those areas in which they are too cold.
**6 Conclusions**
In this paper, we estimate the basal thermal conditions of the Lambert-Amery system
by coupling a forward model and an inverse model, based on six different GHF datasets.
We analyze the contribution of GHF, heat condition, and basal friction to the modelled
basal melt rate. We verify the result using the locations of all known subglacial lakes,
and evaluate the reliability of six GHF datasets in our study domain.
We find significant differences in area with basal melting among the six experiments
due to large variability in GHF. The Li and the Martos fields yield the largest area with
basal melting, and match best with the subglacial lake locations. In contrast, the
Purucker field gives the least area with basal melting and worst match with subglacial
lakes locations. We suggest GHF datasets from Li et al. (2021) and Martos et al. (2017)
as the most suitable choice for this study region.
The modelled high basal friction heating regions are consistent with the fast-flowing
regions. The fast flowing region has smaller modelled basal friction coefficients, and
faster basal velocities. The fast-flowing tributaries have frictional heating in the range
of 50-2000 mW m$^{-2}$. In the vast inland areas, our experiments generally yield high
upward heat conduction in the range of 45-60 mW m$^{-2}$ which means that GHF
dominates the heat content of the basal ice in the slow flow regions. The modelled basal
melt rate reaches 50-500 mm yr$^{-1}$ locally in three very fast flow tributaries (Lambert,
Lepekhin and Kronshtadtskiy glaciers), to the Amery ice shelf, and is in the range of 0-
5 mm yr$^{-1}$ in the inland region.
**Data availability**
All data sets used are publicly available.
**Acknowledgments**
This work was supported by the National Natural Science Foundation of China (No.
41941006) and National Key Research and Development Program of China
(2018YFC1406104).



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
