# Peer review of "Evaluation of six geothermal heat flux maps for the Antarctic Lambert-Amery glacial system"

_The Cryosphere, 2021_

## Referee Comment (RC2)

Kang et al. evaluated the basal thermal conditions of the Lamber-Amery system by using a combined model of a forward model and an inverse model. Results from six experiments based on different geothermal heat flux (GHF) products indicated different distributions of basal temperature and modelled basal melting. By comparing the modelled warm-based region and basal melting rates with locations of subglacial lakes, this study found that two most-recent GHF products based on aerial geomagnetic observations provided best constrain as the basal thermal conditions. Overall the manuscript is generally clearly written. However, the structure needs further modification and some of the description and figures need more improvements.

Here are some general comments:

The finding about consistency between the high basal friction heating and the fast-flowing regions can be easily seen from the way how you calculate the friction heating (Q=tao*velocity), which is less innovative as one of findings in a high-quality peer-reviewed paper.

The section of 3.1.1, 3.1.2, 3.1.3 is nearly same with Wolovick et al. (2021). The authors could just cite this paper rather than copy all these sections. Just make it clear about the different setup you used from Wolovick et al. (2021).

The structure of the paper is a little bit confusing. I suggest moving Sec. 4.2 to Sec. 3.2. Sec. 4.1 Experiment design could fit into end of Sec. 3. Leave Sec. 4.3 as a separate section Sec. 4.

About the improvement of basal friction coefficient, is it original from this study? If yes, I suggest you to mention it in your conclusion section. Besides, I did not see any evaluation about this improvement. Comparison of the difference of simulated and observed surface velocity before and after this improvement is necessary here.

When you talk about the effects of different GHFs on the modelled basal melting, you ignored that fact that different GHFs only affect the modelled basal melting in low-flowing regions even if those six GHFs show different distribution in the fast-flowing region. It further confirmed that friction heating dominated the basal melting for fast-flowing region while the GHF dominated the basal melting in slow-flowing region.

I don't think the Abstract and Conclusions highlight all of the valuable findings in this study. I suggest a serious revision on it.

Several places across the text are lack of citations or need more relevant literature. Some of the figures are not cited accordingly in the text. See the details below.

Specific Comments:

L37: "evidence of extensive subglacial rifts and lakes" citation please.

L77: "for ice temperature" → "ice temperature simulation".

L83-85: Unfinished sentence I guess. "inferred ice and basal temperature"? Or I misunderstood your meaning here.

L101: "in" → "part of"

L104: How did you choose the central streamline here? Where are those datasets (basin boundary, ice front) from? Please add citations.

L115-117: citation of the grounding line dataset and the subglacial lakes.

L123-124: It's not clear tome how and where these two datasets are combined. You should make it clear in Fig. 1.

L156: This is your first time to mention inverse method and Elmer/Ice. Please add citations.

L319: In the boundary condition section (Sec. 3.2.2), you did not mention the constrain for the surface mass balance and basal mass balance for the floating part. Please make it clear here.

L362: This equation is not clearly explained. What is each component in the numerator? Please also add citations for this equation.

L368: The experiment design is quite similar to the multi-cycle spin-up used in Zhao et al. (2018). If yes, please cite the paper here.

L395: citation for the statement "Basal friction in reality depends on basal temperature"

L415: delate "the" after "the modelled".

L416: Do you mean test with different GHFs gave you similar modelled surface velocity? If yes, the statement you made here is not accurate. The only thing you can say is that the inverse method is not sensitive to the choice of GHF product as the boundary condition, which could be one of your findings here.

L427: 500 m/yr. Do you mean the velocity near the GL? If yes, make it clear.

L432: The cyan color is not clear to me. Suggest to change a different color.

L433-434: Why do you chose the contour of 0.5 and -0.5 here? What's the meaning behind those two contours. Please explain.

L339-440: But for the fast-flowing region, we did not see any significant differences. You should make it clear when you talk about the different distribution of warm base.

L441: "In the Li experiment", please cite the figure here. "high" → "highest"

L442: "the basal temperature over most of the domain reaches the melting point", you should add "except for the southern part of domain"

L447: citation for "subglacial mountains"

L455: "heat conduction" → "basal heat conduction". Please add the velocity contour in Fig. 6. About the "fast-flowing tributaries", you didn't define it in Fig. 4a. Do you mean region with velocity higher than 50 m/yr?

L456: "0-30" → "30"

L457-459: Why do you think Purucker shows lower values here? Please explain.

L460: From Fig.7 ,we can tell no significant difference across these 6 experiments. It's better to make a statement here.

L463: when you say reach 2000 mW m-2 at the GL, do you mean all these three glaciers? Or just Lambert?

L478: there are two Fig. 8 here.

L505: I think GHF distribution largely govern basal thermal conditions for the slow-flowing region.

Add citations for "Many previous studies"

L511-L515: Too long sentence. Please split it.

L513: Don't understand what you mean here by "puts warm-based conditions outside of the locations of the observed lakes"

L514: delete "if"

L517: I don't think you use the same inversion method by Wolovick. Do I misunderstand anything here?

L520: What is "ice bed"?

L525: So what? What is the advantage behind it? This could be a highlight of your study.

L542: what do you mean by "ice sheet connected to the ice shelf"? "frictional heating means"? This sentence is not clear to me.

L555: delete ","

L573: "in area" → "in slow flowing area"

---

## Author Comment (AC1)

Referee's comments are in blue, our reply in black, quotes in the revised manuscript in purple.

William Colgan posted a new Referee comment.

**Citation**: https://doi.org/10.5194/tc-2021-357-RC1

This study explores geothermal heat flow in the Lambert-Amery sector of East Antarctica. A complex system of coupled models (shallow ice, full stokes, and subglacial hydrology) are initiated by six available geothermal heat flow maps to estimate basal temperature distribution across the sector. The basal temperatures are generally realistic and conform to expectation, with their differences being informative. Basal heat conduction is also presented and discussed, although the value of this field is less clear to me at the moment. Below, I provide some comments from this article.

Surface Accumulation – The surface accumulation field is used in the balance flux model, and presumably ultimately influences vertical velocity profile. I do not see any description, or citation, documenting the source of the surface accumulation field. It would be helpful to have better description of the accumulation field, including how possible biases in accumulation (or "recent" versus "steady" temporal variations) may manifest in the parameterization of vertical velocity field from the balance flux model, and ultimately in the simulated basal thermal state.

Reply: We note in Section 3: The surface accumulation rate we used in the thermal model was the mean of Arthern et al. (2006) and Van de Berg et al. (2005). Both were accessed through the ALBMAP_v1 dataset (Le Brocq et al., 2010).

We add in the Discussion section of the revision.
We expect that the present-day accumulation rate field will be higher than the long-term average, because of lower accumulation rate during glacial periods. This will tend to increase the downward advection of cold ice in our model, lowering the basal temperature in comparison to reality. On the other hand, we also expect that the modern-day surface temperature will be higher than the long-term average temperature, again because of lower temperatures during glacial periods. This will tend to increase our modeled basal temperature in comparison with reality. It is unclear which of these competing biases is stronger.

References:
Arthern, R. J., Winebrenner, D. P., & Vaughan, D. G. Antarctic snow accumulation mapped using polarization of 4.3-cm wavelength microwave emission. Journal of Geophysical Research: Atmospheres, 111(D6), D06107, 2006. https://doi.org/10.1029/2004JD005667

Le Brocq, A. M., Payne, A. J., & Vieli, A. An improved Antarctic dataset for high resolution numerical ice sheet models (ALBMAP v1). Earth System Science Data, 2(2), 247–260, 2010. https://doi.org/10.5194/essd-2-247-2010

Van de Berg, W. J., Van den Broeke, M. R., Reijmer, C. H., & Van Meijgaard, E. Characteristics of the Antarctic surface mass balance, 1958-2002, using a regional atmospheric climate model. Annals of Glaciology, 41(1), 97-104, 2005. https://doi.org/10.3189/172756405781813302

Topographic Effect – We discuss the topographic effect on geothermal heat flow in https://doi.org/10.1029/2020JF005598. Specifically, we highlight how subglacial topography in the subglacial Gamburtsev Mountains, East Antarctica, can strongly influence geothermal heat flow at kilometer scale. Whereby subglacial ridges can receive 50% less heat flow and subglacial valley can receive 50% more heat flow, in comparison to the regional average. I suspect that explicitly acknowledging the influence of subglacial topography on geothermal heat flow, by applying such a topographic correction field to the GHF input field used by the 2D SIA model, might further improve the realism of simulated hydrology.

Reply:Thanks for your comments. We agree that subglacial topography has influence on geothermal heat at kilometer scale. But we do not know if this effect is positive or negative. It depends on rock type underneath the ice. We had a similar discussion about the influence of subglacial topography on geothermal heat flow in the paper Wolovick et al. (2021b) https://doi. org/10.1029/2020JF005936 as below: "Heat tends to follow the path of least resistance to the surface, so if the thermal conductivity of ice is greater, then heat will be conducted into local valleys and away from local peaks, giving the classic topographic focusing result; but if the thermal conductivity of rock is greater, then the opposite will occur, and heat will tend to be conducted into local peaks and away from valleys (Willcocks & Hasterok, 2019). The thermal conductivity of rock varies with lithology, and can be either greater or less than the thermal conductivity of ice (Willcocks & Hasterok, 2019)."

Therefore, it is unknown how to make the right correction to the GHF input field.

We add these sentences in the Discussion section of the revision: "Subglacial topography has influence on geothermal heat at kilometer scale. Typically, it has been assumed that subglacial ridges receive less heat flow and subglacial valleys receive more heat flow, in comparison to the regional average (e.g., van der Veen et al., 2007; Colgan et al., 2021). However, the effect depends on subglacial rock type. Heat tends to follow the path of least resistance to the surface, i.e. thermal conductivity. The thermal conductivity of rock varies with lithology, and can be either greater or smaller than the thermal conductivity of ice (Willcocks & Hasterok, 2019), thus the sign of topographic effect on GHF can be either negative or positive. Without knowing a priori whether the topographic effect will be positive or negative, it is hard to apply a topographic correction field to the GHF input field."

References:

Wolovick, M. J., Moore, J. C., & Zhao, L. Joint inversion for surface accumulation rate and geothermal heat flow from ice-penetrating radar observations at Dome A, East Antarctica. Part II: Ice sheet state and geophysical analysis. Journal of Geophysical Research: Earth Surface, 126, e2020JF005936, 2021b. https://doi. org/10.1029/2020JF005936

van der Veen, C. J., Leftwich, T., von Frese, R., Csatho, B. M., & Li, J. Subglacial topography and geothermal heat flux: Potential interactions with drainage of the Greenland ice sheet. Geophysical Research Letters, 34(12), 2007.

Colgan, W., MacGregor, J. A., Mankoff, K. D., Haagenson, R., Rajaram, H., Martos, Y. M., et al. Topographic correction of geothermal heat flux in Greenland and Antarctica. Journal of Geophysical Research: Earth Surface, 126, e2020JF005598, 2021. https://doi. org/10.1029/2020JF005598

Willcocks, S., & Hasterok, D. Thermal refraction: Impactions for subglacial heat flux. ASEG Extended Abstracts, 2019(1), 1–4. Taylor & Francis. https://doi.org/10.1080/22020586.2019.12072986

Temperate Ice – I understand how the basal state is parameterized as either melting (Dirichlet; Eq 6) or freeing (Neumann; Eq 5), but the reader would benefit from knowing whether a thicker temperate basal ice layer is permitted. Temperate basal ice layers can form at the convergence of outlet glacier flow, even with tremendous downstream advection relatively cold inland ice (https://doi.org/10.3189/172756502781831322). The presence or absence of a temperate basal ice layer clearly influences the vertical temperature profile, which here seems critical to presented basal heat conduction (i.e. whether temperature gradient simply pinned to the pressure-melting point at the bottom, or the temperature gradient effectively becomes the Clausius–Clapeyron gradient). Allowing ice to become temperate general requires assumptions about liquid pore water content, which I do not see stated here.

Reply: Yes, a thicker temperate basal ice layer is permitted in our model. We add in Section 3.1

In the case that the modelled basal ice temperature reaches pressure melting point, $T_m$, a temperate basal ice layer is permitted in our model. The model works with englacial melting and a temperate ice layer. We do not make assumptions about liquid pore water content. We use a weak-form solution instead of a strict limit. The temperature is allowed to exceed the melting point, but temperature rise is limited by the latent heat absorbed by englacial melting. So, the melt rate rises exponentially as temperature passes the melting point, and the pre-factor for the melt rate comes from the strain heating.

Heat Conduction – I am confused by Figure 8. I would expect that, in the absence of basal hydrologic processes, basal heat conduction is effectively equivalent to geothermal heat flux. Yet inland areas, where basal hydrology is not active, have a very different heat conduction from the forcing geothermal heat flux. The sign of basal heat conduction is also negative, in comparison to the positive sign/direction of geothermal heat flux. Finally, should there not also be "opposing signed" pockets where the basal heat conduction is opposite over subglacial areas where basal water is refreezing (i.e. Vostok in https://doi.org/10.5194/tc-14-4021-2020)? Right now, the axis stops at zero. For these reasons, I find Figure 8 (and associated discussion) difficult to follow.

Reply: The figure about heat conduction is Figure 6. So we assume you are confused by Figure 6 rather than Figure 8 in the previous submission (We add new plots in the revision and the numbering of figures is changed). We checked both figures. Both geothermal heat flux and englacial heat conduction have the same direction, which is upward. It is confusing to use different sign for them.

In the revision, we change "heat conduction" in figure caption of Figure 6 to "modelled heat change of basal ice by upward englacial heat conduction", and add more sentences "The negative sign means that the upward englacial heat conduction causes heat loss from the basal ice as defined by the color bar with cooler colors representing more intense heat loss by conduction."

We note there is a sign typo in Eqn (19), it should be as below

$$M = \frac{G + \vec{u}_b \tau_b + k(T)\frac{dT}{dz}}{\rho_i L}$$

where the term $k(T)\frac{dT}{dz}$ is negative, representing heat loss of basal ice by upward englacial heat conduction.

Our modelled heat change of basal ice by englacial heat conduction is all negative, i.e., there is no places where the basal heat conduction is downward.

The conductive heat flux does not necessarily change sign above subglacial freezing zones. Subglacial freezing happens anywhere that water is available and the basal cooling terms (conduction into the overlying ice sheet and supercooling within the water system) are larger than the basal warming terms (geothermal heat flow, friction heating, and viscous dissipation within the water system). Thus, there is no reason why conduction should change sign at freezing zones; if anything, we would expect conductive cooling to be stronger at freezing zones than at other locations.

We also checked Figure 8. There are three places with negative values of basal melting rate, i.e. refreezing. Therefore, we change Fig. 8 to show modelled freeze-on (see the figure below), and add the text "There are negative values of basal melt rate, which means basal refreezing at three local places (Fig. 9), where there are large gradients in ice thickness typically thinning by 700 m across a distance of 2 km. Radar surveys have not yet been done to confirm these freeze-on locations.".

[Figure]

Fig. Modelled basal melt rate (unit: mm yr$^{-1}$), (a) to (f) correspond to the GHF (a) to (f) in Fig. 2. The ice bottom at pressure-melting point is surrounded by a red contour. The stars denote the locations of observed subglacial lakes, and the area surrounded by the black line is the likely second largest subglacial lake in Antarctica. There is modelled basal refreezing at three local places painted in black.

Subglacial Lakes – Here, subglacial lakes are being used as an indicator of basal ice at the pressure-melting point. The subglacial lake literature, however, now has suggestions that melting temperatures can be depressed significantly lower than pressure-melting point by salinity (https://doi.org/10.1126/sciadv.aar4353) and that radar-derived subglacial water indications can be false positives (https://doi.org/10.5194/tc-14-4495-2020). I think these caveats should be mentioned.

The L511-515 discussion should also use the terminology "false negative" to describe the "one-sided" aspect constraint.

Reply: Agreed, although the Devon Island lake complex is the only known sub-glacial lake that has significantly lowered freezing point temperatures due to dissolved salts, and this is not observed to be the case for Lake Vostok in Antarctica. We add the related discussion as below in the revision. "A lake complex beneath Devon Island ice cap in Canada exists at temperatures well below pressure melting point due to large concentrations of dissolved salts (Rutishauser et al., 2018), and while no similar ones are known to exist beneath the Antarctic ice sheet, direct measurements of ice temperatures above water bodies are rare. Furthermore, relatively high electrical conductivity beds such as water saturated clays can give rise to false positives in radar detections of subglacial water bodies (Talalay et al., 2020)."

References:

Rutishauser A., Blankenship, D. D., Sharp, M., Skidmore, M. L., Greenbaum, J. S., Grima, C., Schroeder, D. M., Dowdeswell, J. A., Young, D. A., Discovery of a hypersaline subglacial lake complex beneath Devon Ice Cap, Canadian Arctic, Sci. Adv.2018; 4: eaar4353

Talalay, P., Li, Y., Augustin, L., Clow, G. D., Hong, J., Lefebvre, E., Markov, A., Motoyama, H. and Ritz, C., Geothermal heat flux from measured temperature profiles in deep ice boreholes in Antarctica, The Cryosphere, 14, 4021-4037, 2020

Consensus Map – Given the time and interest that the authors have clearly expended with this study, it would seem that they are in a very good position to produce a consensus geothermal heat flow map of the Lambert-Amery sector. My final thought would be asking why the authors simply stop with saying the Li and Martos geothermal heat flow maps are most suitable for this region, and do not provide an accompanying data product of a geothermal heat flow map that is self-consistent, or optimized, with an ice flow model (i.e. https://doi.org/10.20575/00000006)?

Reply: Unfortunately, we don't think we can do this. We note in the Conclusions: We cannot make our own GHF map from our analysis since while we can pick the GHF where Li and Martos geothermal heat flow maps are consistent and both agree with the observation, we do not know which (if either) are correct where the Li and Martos GHF datasets disagree and there are no observations. We would need additional observations of measured basal temperature from deep ice cores, or observed refreeze-on, but neither are available in the region.

We read your suggested reference (Greve, 2019). Ralf Greve presented an improved distribution of the geothermal heat flux for Greenland. He did a paleoclimatic simulation carried out with the ice sheet model SICOPOLIS, and modified the GHF

values at five deep ice core locations such that observed and simulated basal temperatures match closely. However, there is no deep ice core drilling site in our study region.

References:

Greve R., Geothermal heat flux distribution for the Greenland ice sheet, derived by combining a global representation and information from deep ice cores, Polar Data Journal, 3, 22–36, 2019

---

## Author Comment (AC2)

Referee's comments are in blue, our reply in black, quotes in the revised manuscript in purple.

Kang et al. evaluated the basal thermal conditions of the Lamber-Amery system by using a combined model of a forward model and an inverse model. Results from six experiments based on different geothermal heat flux (GHF) products indicated different distributions of basal temperature and modelled basal melting. By comparing the modelled warm-based region and basal melting rates with locations of subglacial lakes, this study found that two most-recent GHF products based on aerial geomagnetic observations provided best constrain as the basal thermal conditions. Overall the manuscript is generally clearly written. However, the structure needs further modification and some of the description and figures need more improvements.

Here are some general comments:

The finding about consistency between the high basal friction heating and the fast-flowing regions can be easily seen from the way how you calculate the friction heating (Q=tao*velocity), which is less innovative as one of findings in a high-quality peer-reviewed paper.
Reply: Agreed. We remove the sentences in the abstract and conclusion talking about the consistency between the high basal friction heating and the fast-flowing regions.

The section of 3.1.1, 3.1.2, 3.1.3 is nearly same with Wolovick et al. (2021). The authors could just cite this paper rather than copy all these sections. Just make it clear about the different setup you used from Wolovick et al. (2021).
Reply: We removed most words and the separate Section of 3.1.1, 3.1.2, 3.1.3, and pointed the different setup we used from Wolovick et al. (2021), which is mainly about how we used a merged surface flow direction field, and how we use basal sliding ratio computed by the full-Stokes inverse model to constrain rheology and shape function model in the forward model.

The structure of the paper is a little bit confusing. I suggest moving Sec. 4.2 to Sec. 3.2. Sec. 4.1 Experiment design could fit into end of Sec. 3. Leave Sec. 4.3 as a separate section Sec. 4.
Reply: Thank you for the suggestions. We modified the structure of the paper as you suggested. Due to the changes in text, we also reordered the number of equations.

About the improvement of basal friction coefficient, is it original from this study? If yes, I suggest you to mention it in your conclusion section. Besides, I did not see any evaluation about this improvement. Comparison of the difference of simulated and observed surface velocity before and after this improvement is necessary here.
Reply: Yes, the improvement of basal friction coefficient is original from this study. We mention it in our conclusion section of the revision. We also improve the basal friction calculation to include information on the basal ice temperature relative to its pressure

melting point. This procedure results in removal of unrealistic noise manifested as local spikes in modelled basal friction heat.

The goal of this improvement is to remove unrealistic noise manifested as local spikes in modelled basal friction heat. We show two comparison plots below, one shows the comparison of modelled basal friction heat before and after this improvement, the other shows the difference of simulated and observed surface velocity before and after this improvement.

We can see that the unrealistic noise is much less after this improvement, and the difference of simulated and observed surface velocity is unchanged in the region except for some parts of the inland boundary.

[Figure]

Figure: Comparison of modelled basal friction heat with basal friction coefficient $\beta_{old}$ (a); and $\beta_{new}$ with $\alpha=1$ (b). The white square is enlarged.

[Figure]

This figure (not in the manuscript) shows the difference between simulated and observed surface velocity plotted as $\log_{10}$(modeled/observed) using different basal friction coefficients $\beta_{old}$ (a); and $\beta_{new}$ with $\alpha=1$ (b). The white lines in represent contours of 0.5 (a ratio of modeled/observed of about 3) and the black lines represent contours of -0.5 (a ratio of about 1/3).

When you talk about the effects of different GHFs on the modelled basal melting, you ignored that fact that different GHFs only affect the modelled basal melting in low-flowing regions even if those six GHFs show different distribution in the fast-flowing region. It further confirmed that friction heating dominated the basal melting for fastflowing region while the GHF dominated the basal melting in slow-flowing region.

Reply: We do not fully agree with your opinion that "different GHFs only affect the modelled basal melting in low-flowing regions". GHFs not only affect the extent of basal melting but also affect the magnitude of basal melting rate. Although there is basal melt in fast-flowing region using different GHFs, the magnitudes of basal melt rates are different. For instance, use of Purucker GHF which is lower than other GHFs in the fast flow region produces smaller basal melt rate in the fast-flowing region. We added: The fast-flowing region has smaller modelled basal friction coefficients, and faster basal velocities, but there are large differences in basal melting rates between the 6 GHF datasets.

I don't think the Abstract and Conclusions highlight all of the valuable findings in this study. I suggest a serious revision on it.

Reply: We revised the Abstract and Conclusions in the revision.

Abstract:

Basal thermal conditions play an important role in ice sheet dynamics, and they are sensitive to geothermal heat flux (GHF). Here we estimate the basal thermal conditions, including basal temperature, basal melt rate, and friction heat underneath the Lambert-Amery glacier system in east Antarctica, using a combination of a forward model and an inversion from a 3D ice flow model. We assess the sensitivity and uncertainty of basal thermal conditions using six different GHFs. We evaluate the modelled results using all observed subglacial lakes. The different GHFs lead to large differences in simulated spatial patterns of temperate basal conditions. The two recent GHF fields inverted from aerial geomagnetic observations have the highest GHF, produce the largest warm-based area, and match the observed distribution of subglacial lakes better than the other GHFs. The modelled basal melt rate reaches ten to hundreds of mm per year locally in Lambert, Lepekhin and Kronshtadtskiy glaciers feeding the Amery ice shelf, and ranges from 0-5 mm yr$^{-1}$ on the temperate base of the vast inland region.

**Conclusions**

In this paper, we estimate the basal thermal conditions of the Lambert-Amery system by coupling a forward model and an inverse model, based on six different GHF datasets. We analyze the contribution of GHF, heat conduction, and basal friction to the modelled basal melt rate. We verify the result using the locations of all known subglacial lakes, and evaluate the reliability of six GHF datasets in our study domain.

Our approach is distinct from that used to find GHF fields employed by Wolovick et al. (2021a), in particular the use of a full Stokes model allows the method to be extended to fast flowing ice stream and ice shelf domains where neither the shallow ice or shallow shelf-approximations are valid. We also improve the basal friction calculation to include information on the basal ice temperature relative to its pressure melting point. This procedure results in removal of unrealistic noise manifested as local spikes in modelled basal friction heat.

We find significant differences in the spatial extent of temperate ice in the slow flowing areas among the six experiments due to large variability in GHF. The experiments using Li et al. (2021) and the Martos et al. (2017) GHF yield the largest area with basal melting, and match the subglacial lake locations best. In contrast, the experiments using Purucker (2013) GHF gives the least area with basal melting and the worst match with subglacial lakes locations. We suggest GHF datasets from Li et al. (2021) and Martos et al. (2017) as the most suitable choice for this study region. We cannot make our own GHF map from our analysis since while we can pick the GHF where Li and Martos geothermal heat flow maps are consistent and both agree with the observations, we do not know which (if either) are correct where the Li and Martos GHF datasets disagree and there are no observations. In order to make this determination we would need additional observational constraints on the basal thermal state, such as measured basal temperatures from deep ice cores, or observed refreeze-on, but neither are available in the region.

The fast-flowing region has smaller modelled basal friction coefficients, and faster basal velocities, but there are large differences in basal melting rates between the 6 GHF datasets. The fast-flowing tributaries have frictional heating in the range of 50-2000 mW m$^{-2}$. In the vast inland areas, our experiments generally yield high upward heat conduction in the range of 45-60 mW m$^{-2}$ which means that GHF dominates the heat content of the basal ice in the slow flow regions. The modelled basal melt rate reaches 50-500 mm yr$^{-1}$ locally in three very fast flow tributaries (Lambert, Lepekhin and Kronshtadtskiy glaciers) feeding the Amery ice shelf, and is in the range of 0-5 mm yr$^{-1}$ in the inland region.

Several places across the text are lack of citations or need more relevant literature. Some of the figures are not cited accordingly in the text. See the details below.

Specific Comments:
L37: "evidence of extensive subglacial rifts and lakes" citation please.
Reply: We add references.
L77: "for ice temperature" → "ice temperature simulation".
Reply: done.
L83-85: Unfinished sentence I guess. "inferred ice and basal temperature"? Or I misunderstood your meaning here.
Reply: We change this sentence "Large scale studies on the dependence on GHF of the Greenland (Rezvanbehbahani et al., 2019) and Antarctica ice sheet (Pattyn, 2010) have inferred ice and basal temperatures" to "Glaciologists have combined ice sheet models with measurements of vertical temperature or thawed basal state to constrain GHF of the ice sheets (e.g. Pattyn, 2010; Rezvanbehbahani et al., 2019)".
L101: "in" → "part of"
Reply: Done.
L104: How did you choose the central streamline here? Where are those datasets (basin boundary, ice front) from? Please add citations.
Reply: We made it ourselves. The central streamline was chosen by selecting a point at

the confluence of Lambert Glacier and Lepekhin Glaicer and then advecting that point downstream to the ice front using the observed velocity field.

L115-117: citation of the grounding line dataset and the subglacial lakes.

Reply: We add the citations. "The red curve is part of the grounding line of Amery ice shelf (Morlighem et al., 2020) … The black stars in (c) denote the locations of observed subglacial lakes (Wright and Siegert, 2012; Cui et al., 2021)"

L123-124: It's not clear to me how and where these two datasets are combined. You should make it clear in Fig. 1.

Reply: In Fig. 1, we add a dotted red curve in plot (b) showing the boundary of ice thickness data from Cui et al. (2020a). We use the data from Cui et al. (2020a) inside this boundary and BedMachine data outside this boundary.

L156: This is your first time to mention inverse method and Elmer/Ice. Please add citations.

Reply: Done.

L319: In the boundary condition section (Sec. 3.2.2), you did not mention the constrain for the surface mass balance and basal mass balance for the floating part. Please make it clear here.

Reply: In Elmer/Ice model, we do diagnostic simulation, i.e., we perform a stress-balance snapshot. Therefore, we do not need to prescribe surface mass balance or basal mass balance in the boundary conditions for the ice sheet including the ice shelf. We add the explanation in section 3.2.2.

L362: This equation is not clearly explained. What is each component in the numerator? Please also add citations for this equation.

Reply: There was a typo in Eq (19). We corrected it in the revision.

$$M = \frac{G + \vec{u}_b \tau_b + k(T)\frac{dT}{dz}}{\rho_i L}$$

where M is the basal melt rate, G is GHF, $\vec{u}_b \tau_b$ is the basal friction heat, $-k(T)\frac{dT}{dz}$ is the upward heat conduction, $n$ is the outward unit vector at the ice bottom, $\rho_i$ is the ice density, and $L$ is latent heat of ice melt. We add the reference for this equation:

Greve R, Blatter H, Dynamics of Ice Sheets and Glaciers, Springer, 2009.

L368: The experiment design is quite similar to the multi-cycle spin-up used in Zhao et al. (2018). If yes, please cite the paper here.

Reply: It is similar. We cite the paper Zhao et al. (2018).

L395: citation for the statement "Basal friction in reality depends on basal temperature"

Reply: We add the reference:

Greve R, Blatter H, Dynamics of Ice Sheets and Glaciers, Springer, 2009.

L415: delate "the" after "the modelled".

Reply: Done.

L416: Do you mean test with different GHFs gave you similar modelled surface velocity? If yes, the statement you made here is not accurate. The only thing you can say is that

the inverse method is not sensitive to the choice of GHF product as the boundary condition, which could be one of your findings here.

Reply: The inverse method is designed to minimize the misfit between modelled and observed surface velocity. Therefore, it is not surprising that the modelled surface velocities are similar for the different GHFs. This is not a finding. It is what one expected.

We change "In the inverse method, the modeled surface velocity matches best to the observed surface velocity. Therefore, we get very similar distributions of modeled velocity field using different GHFs" to "In the inverse method, the misfit between the modeled and the observed surface velocity is minimized. Therefore, we get very similar distributions of modeled surface velocity field using different GHFs.".

L427: 500 m/yr. Do you mean the velocity near the GL? If yes, make it clear.

Reply: Yes, done.

L432: The cyan color is not clear to me. Suggest to change a different color.

Reply: We update Fig. 4 in the revision. We use white solid lines in (a), (b), and (d) to plot speed contours of 50, 100 and 200 m yr$^{-1}$.

L433-434: Why do you chose the contour of 0.5 and -0.5 here? What's the meaning behind those two contours. Please explain.

Reply: We do not think this subplot is helpful, so we remove it in the revision, just using it reply to your earlier general comment above.

The values are arbitrary and simply show the ranges of the velocity differences. The

contour 0.5 means $\dfrac{\text{modelled velocity}}{\text{observed velocity}} = 10^{1/2} \approx 3.1$ , and the contour -0.5 means

$\dfrac{\text{modelled velocity}}{\text{observed velocity}} = 10^{-1/2} \approx \dfrac{1}{3}$. We use ratio of 1/3~3 times to compare the difference of

modelled and observed velocity. Modelled velocity in most region is in this range.

L339-440: But for the fast-flowing region, we did not see any significant differences. You should make it clear when you talk about the different distribution of warm base.

Reply: We assume you mean L439-440. We modify it to "The modelled ice bottom of fast-flowing region are all warm based (basal temperature reaching the pressure melting point). However, there are significant differences in the modelled distribution of warm base in the slow-flowing region using different GHFs."

L441: "In the Li experiment", please cite the figure here. "high" → "highest"

Reply: Done.

L442: "the basal temperature over most of the domain reaches the melting point", you should add "except for the southern part of domain"

Reply: Done.

L447: citation for "subglacial mountains"

Reply: Subglacial mountains are shown in Fig. 1c. so we refer to Fig. 1c here.

L455: "heat conduction" → "basal heat conduction". Please add the velocity contour in Fig. 6. About the "fast-flowing tributaries", you didn't define it in Fig. 4a. Do you mean region with velocity higher than 50 m/yr?

Reply: To be more clear, we changed "heat conduction" to "modelled heat change of basal ice by upward englacial heat conduction".

The fast-flowing tributaries, we mean the region with velocity higher than 30 m/yr. We add velocity contours of 30, 50, 100, 200 m/yr in Fig. 6 (which is Fig. 8 in the revision).

L456: "0-30" → "30"

Reply: Done.

L457-459: Why do you think Purucker shows lower values here? Please explain.

Reply: This sentence is only for vast inland areas (slow-flowing region). We can tell that from the colorbar in Fig. 6. Purucker (Fig. 6e) has lighter color than other subplots.

L460: From Fig. 7, we can tell no significant difference across these 6 experiments. It's better to make a statement here.

Reply: We add a statement here "There is no significant difference in modelled basal friction heat across these 6 experiments."

L463: when you say reach 2000 mW m$^{-2}$ at the GL, do you mean all these three glaciers? Or just Lambert?

Reply: We change it to "The three fast-flowing tributaries have friction heat amounting to more than 50 mW m$^{-2}$, with the Lambert and Kronshtadtskiy glaciers having 2000 mW m$^{-2}$ at the grounding line."

L478: there are two Fig. 8 here.

Reply: we remove one.

L505: I think GHF distribution largely govern basal thermal conditions for the slow-flowing region. Add citations for "Many previous studies"

Reply: Done. We add citations Larour et al., 2012; Pattyn, 2010; Pittard et al.,2016; Van Liefferinge and Pattyn, 2013; Van Liefferinge et al. 2018.

L511-L515: Too long sentence. Please split it.

Reply: we change this sentence to "However, it should be noted that observations of subglacial lakes are a one-sided constraint. A model result that misses the observed lakes is clearly too cold at that location. But if the model result shows basal melt at a place with no observed lakes, it is not clear whether this is because the model is too warm, or if the subglacial water exists in a form other than in ponded lakes.".

L513: Don't understand what you mean here by "puts warm-based conditions outside of the locations of the observed lakes"

Reply: We mean "if the model result shows basal melt at a place with no observed lakes", see the above reply.

L514: delete "if"

Reply: see the above reply.

L517: I don't think you use the same inversion method by Wolovick. Do I misunderstand anything here?

Reply: That is correct, we do not use the same inversion method as Wolovick et al (2021). That paper adjusted GHF and surface accumulation rate to fit observations of subglacial lakes, basal freeze-on, and internal layers. We only use the forward model described in that paper for our thermal and hydrology model. The inverse model used here, by contrast, is a classical ice dynamic inversion that adjusts basal friction to match surface velocity.

We change this sentence "Our methodology builds on the earlier inversion method employed by Wolovick et al. (2021)" to "Our approach is distinct from that used to find GHF fields employed by Wolovick et al. (2021a), in particular the use of a full Stokes model allows the method to be extended to fast flowing ice stream and ice shelf domains where neither the shallow ice nor shallow shelf-approximations are valid."

L520: What is "ice bed"?

Reply: we change it to "ice bottom".

L525: So what? What is the advantage behind it? This could be a highlight of your study.

Reply: We mention this in the conclusions: We also improve the basal friction calculation to include information on the basal ice temperature relative to its pressure melting point. This procedure results in removal of unrealistic noise manifested as local spikes in modelled basal friction heat.

The goal of this improvement of $\beta$ is to reduce the local spikes in modelled friction heat. The modelled surface velocity after the improvement of $\beta$ is unchanged in the region except for some parts of the inland boundary.

L542: what do you mean by "ice sheet connected to the ice shelf"? "frictional heating means"? This sentence is not clear to me.

Reply: Sorry that we did not express clearly. It means "grounded ice sheet near the ice shelf". We change it to "Most GHF distributions (except Martos et al., 2017 and Li et al., 2021) in the grounded ice sheet near the ice shelf are homogeneous, but frictional heating in the fast-flowing ice is more than 10 times higher than in the slow-flowing ice."

L555: delete ","

Reply: done.

L573: "in area" →"in slow flowing area

Reply: done.

---

## Author Comment (AC3)

Referee's comments are in blue, our reply in black, quotes in the revised manuscript in purple.

This paper describes a method that can be used to evaluate observations of geothermal heat flux, and its application to a large region of Antarctica: the Lambert glacier and its drainage basin. A sophisticated model is used to estimate warm bedded regions given observations of ice geometry and surface velocity and several estimates of the geothermal heat flux field (GHF). The resulting warm bedded regions vary considerably depending on the choice of GHF, allowing the paper to rank them by comparing those regions to known locations of sub-glacial lakes. The Lambert glacier is representative of the majority of Antarctica since it is large and includes cold based ice, warm based ice that sees slow sliding, and warm based ice that sees fast sliding. That suggest that the method could be applied more widely, so both the method and its results should be of interest. The paper is generally well written and clear.

Thanks for your encouraging comments.

**General Comments**

The ice flow model (a Stokes flow model) is a complex one. It is certainly a better choice given unlimited resources than any of its common approximations (SIA, SSA, HOM…) and looks to have been applied correctly, but why is it necessary in this case? In which parts of the domain? It seems that in some parts you only use the direction from Elmer/Ice: how much does that differ from the direction of the surface gradient? The discussion says that this work 'builds on the earlier inversion method employed by Wolovick et al'. (which is SIA based) but how important is that extra effort?

Reply: Unlike Wolovick et al. (2021), we can not use SIA because we have a fast-flowing glacier and a floating shelf in our domain, and SIA does not represent those regimes well. Conversely, we cannot use SSA because we would also like to have an accurate solution in slow-flowing interior areas that move by internal deformation, and SSA does not represent internal deformation. We could have used a Higher Order model instead of full stokes, however, Elmer/Ice does not have an option for HOM. Therefore, our only option was full stokes.

Our description is incomplete. The surface velocity actually has 3 sources: the direction of surface gradient, Elmer/Ice modelled velocity and observations. The observations are used where flow is fast, Elmer/Ice modelled velocity is used where flow is slow, and the surface gradient is only used near the margins of the domain where the Elmer/Ice modelled velocity is not reliable. We add these descriptions in the revision.

We compared the direction from surface gradient, Elmer/Ice modelled velocity, and the observed velocity direction, see the figure below. As shown, there is large difference between modelled and observed velocity in the slow flow region. The Elmer/Ice model gives a better velocity field, and it is important because we need to use the modelled

basal velocity and basal shear force to calculate the basal friction heat and basal melting rate. We add this plot in section 3.3 of the revision.

[Figure]

Figure. Velocity direction fields, in degrees clockwise from grid north. The first row shows the direction from surface gradient, Elmer/Ice modelled velocity, and the observed velocity direction. The middle row shows the 3 weighting fields (the sum of these weights is 1). The bottom row shows the difference between the direction of surface gradient and Elmer/Ice modelled velocity (plot g), the difference between the observed velocity direction and Elmer/Ice modelled velocity (plot h), and the merged velocity field used in the forward model (plot i).

Equation 4 appears to be a 'cold ice' model, i.e one that assumes T < Tm. The manuscript should justify this choice, with reference to polythermal models e.g Aschwanden 2013.

Reply: Yes, the modelled ice temperature is subject to the condition $T \le T_m$. Another referee said "The section of 3.1.1, 3.1.2, 3.1.3 is nearly same with Wolovick et al. (2021). The authors could just cite this paper rather than copy all these sections. Just make it clear about the different setup you used from Wolovick et al. (2021)." Therefore, we removed the descriptive text on the same setup as used from Wolovick et al. (2021) in the revision, including this Equation 4.

Use consistent notation for vectors etc throughout.

Reply: Done.

**Specific Comments**

L17. Are abbreviations (GHF) permitted in the abstract?

Reply: We think so but we are not sure. We can change if the editor says no.

L38. "Suggesting.."? How?

Reply: Rewritten as: However, there is also evidence of extensive subglacial rifts and lakes (Fretwell et al., 2013; Jamieson et al., 2016; Cui et al., 2020a). Jamieson et al. (2016) report a large subglacial drainage network in Princess Elizabeth Land (PEL), which would transport water from central PEL toward the Lambert-Amery region. The complexity of subglacial environment may influence the stability and basal mass balance of this area.

L41 ice penetrating radar *data*

Reply:Done.

L50, infers -> implies?

Reply: Done.

L67. comments on melt-water routing seem out of place in this paragraph

Reply: We move this sentence to an earlier location in this paragraph. Then it is "Ice at the melting point can lead to water, flowing along hydraulic gradients, and accumulating in local depressions (Fricker et al., 2016). The meltwater lubricates the ice/bed interface or saturates any sediment till layer and facilitates higher ice velocities via basal sliding."

L73. 'Ice sheet models are useful tools' is a matter of opinion, and not connected to the rest of the paragraph.

Reply: We change it to "Ice sheet models can be used to simulate the dynamics and thermodynamics of the ice sheet", and move it to the beginning of the next paragraph.

L97 "Hence, we make inferences" –> We state / We determine?

Reply: Disagreed. We think "we make inferences on …" is correct usage here.

Section 2

L103 and fig2 –rephrase, and draw the whole shelf/gl so that the reader can easily tell what is meant by 'half'. Related to this, in 3.3.3 explicitly state the boundary condition at this segment of lateral boundary (I assume it is the same as the other boundaries) and give a justification.

Reply: We assume you mean to draw the whole shelf/gl in Fig. 1 rather than Fig. 2, because Fig. 1 is included in our sentence "It consists of two drainage basins: the Lambert Glacier Basin, the American Highland Basin, along with about half of Amery Ice Shelf (Fig. 1)."

In the updated Fig. 1, we draw the whole grounding line of Amery ice shelf, we added "about" in the description along with about half of Amery.

[Figure]

The updated Fig. 1 in the revision. The domain topography and location with domain boundary overlain. (a) surface elevation; (b) ice thickness; (c) bed elevation; (d) the location of our domain in Antarctica. The solid black curve is the outline of the study domain, including the central streamline of Amery ice shelf and the boundary of inland sub-basins based on drainage-basin boundaries defined from satellite ice sheet surface elevation and velocities (Mouginot et al., 2017; Rignot et al., 2019). The solid red curve is the grounding line of Amery ice shelf (Morlighem et al., 2020). The dotted black curve is the dividing line between Lambert Glacier Basin and the American Highland Basin. The dotted red curve in (b) is the boundary of ice thickness data from Cui et al. (2020a). The black stars in (c) denote the locations of observed subglacial lakes (Wright and Siegert, 2012; Cui et al., 2021), the region within the black line at (1800E, 300N) is potentially the second largest subglacial lake in Antarctica. The red arrows in (c) indicate the routing through the deep subglacial canyon system from GSM to WIS.

L:156 'Inverse method'- no such thing. You are solving an inverse (that is, ill-posed) problem, using (most likely) some sort of gradient based optimization method. You are also not estimating ice flow velocity and stress, but inferring the basal friction such that the model velocity best fits observations.

Reply: We change the sentence in L156 to "We solve an inverse problem by a full-Stokes model, implemented in Elmer/Ice, to infer the basal friction coefficient such that the model velocity best fits observations (Gagliardini et al., 2013). Using the best-fit basal friction coefficient, we obtain the ice flow velocity, stress, and basal friction heat."

We also change "inverse method" elsewhere to "inverse problem" or "inverse model".

L158-163 – Some rewording is needed here. You don't describe the procedure that you hint at for some time, so provide a summary here ('we will describe each model component in sections X and Y, then the coupling in Z')

Reply: Thanks for your comments. We adjust the structure and provide a summary here. "We will describe the forward model in Section 3.1 and the inverse model in Section 3.2, then the coupling in Section 3.3."

L223 In general, this section need to be cleared up, how for example does 'water input supply a large freezing rate'.

Reply: Most of this section has been removed, since another referee said we could just cite this paper rather than copy all these sections, and we just discuss the differences in setup from Wolovick et al. (2021).

L219: no need to say 'taking six GHF datasets…' or at least rephrase to be clear that you only use one at a time.

Reply: This sentence is removed.

L329; Eq 15 is not the Weertman law, it is a linear viscous law which works satisfactorily in inverse problems (because you are really finding Tb, not C) but not in general.

Reply: We change "Weertman law" to "a linear sliding law".

L347 Use subscripts consistently

Reply: we change $u^{obs}$ to $u_{obs}$.

L363 (and elsewhere) the conductive heat flux Fc = -k dT/dz is positive (upward) when the bed is warmer than the ice above, so should you not have + k dT/dz (i.e – Fc ) if the bed "loses heat from upward heat conduction" . What about the case where dT/dz is negative (pressure melting point reached above the ice bed). Does that simply never happen?

Reply: Yes, we note there is a sign typo in this equation, it should be as below

$$M = \frac{G + \vec{u}_b \tau_b + k(T)\frac{dT}{dz}}{\rho_i L}$$

where the term $k(T)\frac{dT}{dz}$ is negative, representing heat loss of basal ice by upward englacial heat conduction.

We add this paragraph in the revision: "In the case that the modelled basal ice temperature reaches pressure melting point, $T_m$, a temperate basal ice layer is permitted in our model. The model works with englacial melting and a temperate ice layer. We do not make assumptions about liquid pore water content. We use a weak-form solution instead of a strict limit. The temperature is allowed to exceed the melting point, but temperature rise is limited by the latent heat absorbed by englacial melting. So, the melt rate rises exponentially as temperature passes the melting point, and the pre-factor for the melt rate comes from the strain heating. "

L400. This procedure seems important but is glossed over. If Bnew != Bold, then why does the modelled surface velocity not change?

Reply: We note in the text "the difference of simulated and observed surface velocity is unchanged in the whole region except for some parts of the inland boundary." We note in the conclusions "We also improve the basal friction calculation to include information on the basal ice temperature relative to its pressure melting point. This procedure results in removal of unrealistic noise manifested as local spikes in modelled basal friction heat. "

We add a figure as below to compare modelled basal friction heat before and after this improvement of $\beta$.

[Figure]

Figure: Comparison of modelled basal friction heat with basal friction coefficient $\beta_{old}$ (a) and $\beta_{new}$ with $\alpha=1$ (b). The white square is enlarged.

L450 and fig 6. Why is the heat flux negative? Especially since you talk about magnitudes in the text.

Reply: In the revision, we change "heat conduction" in figure caption of Figure 6 to "modelled heat change of basal ice by upward englacial heat conduction", and add more sentences "The negative sign means that the upward englacial heat conduction causes heat loss from the basal ice as defined by the color bar with cooler colors representing more intense heat loss by conduction."

We also change the text correspondingly.

L460 and fig 7 – it is difficult to tell the difference between these. Would it help to show differences relative to (a) Martos? That said, you don't seem to depend much on these figures so are they really needed?

Reply: Agreed. We can see no significant difference across these 6 experiments, We add a new plot (the new Fig. 4) in the revision showing the modeled basal friction heat before and after the improvement of $\beta$ using Martos GHF. So we do not need the old plot Fig. 7 in the revision. We change in the revision "There is no significant difference in modelled basal friction heat across these 6 experiments. We only show the modelled basal friction in experiment using Martos et al. (2017) GHF (Fig. 4b)."

L487. It is not quite accurate to say that 'The Li experiment gives the best fit' (how is the fit quantified?). I suggest rephrasing along the lines of the following sentence which sums up the results more accurately, i.e it is only the Li experiment that results in a warm base that covers all observed lakes.

Reply: We rephrased these sentences as you suggested "The modelled warm base in the experiment using Li et al. (2021) GHF covers all the observed subglacial lakes in the domain, including the recently discovered second-largest subglacial lake in Antarctica (Cui et al., 2020b). The warm base in the experiment using Martos et al. (2017) GHF covers the second most observed subglacial lakes, and the experiment using An et al. (2015) GHF the third".

L506-507 (and elsewhere): The datasets/fields should be referenced correctly '(Li et al 2021)', rather than just 'Li'. I also would prefer to see you write 'our experiment using the Li et al 2021 GHF' rather than 'the Li experiment', but I don't think there is any real danger of the reader being misled by that.

Reply: We change "Li" to 'Li et al. (2021)', and similar for other GHF datasets. We also change 'the Li experiment' to 'experiment using the Li et al. (2021) GHF', and similar change for other GHF datasets all through the text.

---

## Editor Decision (ED1)

**Editor's comments**

I would like to thank the authors for submitting a revised version of their manuscript and for responding to all reviewer and editor comments. Several sections of the manuscript have been significantly re-structured, and this has greatly improved the clarity of the paper – thank you for responding to this suggestion from one of the reviewers and for seamlessly implementing the edits.

I am grateful to two of the reviewers who provided feedback on the resubmitted manuscript. One reviewer has requested a couple of minor edits, and both are happy that the article is now suitable for publication.

I have read through the revised version of the article and list below a number of points that require clarification prior to publication – please get in touch if anything is unclear. The points are all minor and therefore I recommend this article can now be 'published subject to technical corrections' (i.e. it does not need to undergo any further review by the editor or reviewers).

Thank you for choosing to publish your work in The Cryosphere.

Kind regards,

Pippa Whitehouse (editor)

**Minor line-by-line comments**

Line 20: "use of six different GHFs" – not clear if this refers to six different GHF models, or simply six different GHF values, please clarify and review use of this phrase throughout the manuscript

Line 32: does "subglacial rifts" refer to a geological feature or a glaciological feature? If the former, it is not clear how this is relevant when discussing the stability of the Lambert-Amery system on a decadal timescale – please clarify what type of feature you are referring to and how it is relevant

Line 79: "…the basal thermal conditions inferred from the new high-resolution topography dataset" – include a reference to clarify which study you are referring to

Line 105: suggest "The margins of the inland sub-basins…"

Figure 1: it would be useful to depict the coastline/grounding line of Antarctica in plots (a)-(c). Also, I suggest adding text labels to plot (b) to clarify which domain uses data from Cui et al. (2020) and which domain uses data from MEaSUREs – one reviewer comments that this is still unclear

Line 165: clarify what you mean by a 'proper' initial temperature

Line 170: clarify which models you are referring to, e.g. "coupling the forward and inverse models"

Line 179: "downhill in the ice surface" – unusual phrasing, perhaps 'along flowlines'?

Line 185: 'step' -> 'component' (using the terminology from line 178)

Equation 1: define 'm' and k(T), perhaps also stating whether they take positive or negative values

Fig. 3 caption: specify (here and elsewhere) whether this is surface velocity or the full velocity field

Line 227: in what way is the basal slip ratio 'added' to the method? If the method already uses a basal slip ratio perhaps the novel feature here is that you use a spatially variable basal slip ratio?

Equations 3 and 4: suggest using the del/nabla symbol when representing div, grad etc.

Line 287: on line 275 you state that beta is the basal friction coefficient, check use of terminology

Line 299: here and elsewhere, you could replace 'do', 'done', 'did' with 'carry out' or 'carried out' when referring to the methods used, e.g. "An L-curve analysis has been carried out to find…"

Line 312: "…from by…" – typo

Line 312: is $T_M$ the same as $T_m$ (defined on line 202)?

Figure 5: state that this figure shows results for the Martos et al. (2017) model

Lines 331-332: could relate these statements about heat to the impact of each term on the basal melt rate – this would help to clarify the sign of the final term in the numerator of eq. 10

Line 335: similar as -> similar to

Line 336: refer to figure 3 when referring to the 'mixture' of information used to determine the input surface velocity (also on line 345 when discussing merging three different pieces of information)

Line 347: by -> from

Line 438: figure 9?

Line 442-443: suggest "Regions with positive basal melt rates…"

Line 454-455: suggest "…the second greatest number of observed subglacial lakes…"

Line 477: suggest "the …accumulation rate field used in our modelling will be…" (similar on line 181)

Line 478: "because of lower accumulation rates during glacial periods" – reference needed

Line 507: suggest "…that does not predict basal melt at the location of the observed lakes…"

Line 509: a third possibility is that lakes are present, but we do not have the data to detect them

Line 515: "direct measurements… are rare" – the important point here is that lakes can exist where basal temperatures are below the pressure melting point, suggest revising the text to reflect this

Line 517: the paper suggested by the reviewer 1 is by Tulaczyk et al. (2020) not Talalay et al. (2020) – see https://tc.copernicus.org/articles/14/4495/2020/

Line 530: Antarctica -> Antarctic

Line 533: "…to the our…" – typo

Line 557: include full references for the Martos and Li GHF models (also lines 592-594)

line 600: "…has smaller modelled basal friction coefficients…" – is this information quantified anywhere in the manuscript?

Please carry out a careful check for grammatical issues, particularly in relation to the use of singular and plural and the definite article ('the'), which is occasionally missing

---

## Author Response (AR2)

Editor's comments are in black, our reply in blue.

**Editor's comments**

I would like to thank the authors for submitting a revised version of their manuscript and for responding to all reviewer and editor comments. Several sections of the manuscript have been significantly re-structured, and this has greatly improved the clarity of the paper – thank you for responding to this suggestion from one of the reviewers and for seamlessly implementing the edits.

I am grateful to two of the reviewers who provided feedback on the resubmitted manuscript. One reviewer has requested a couple of minor edits, and both are happy that the article is now suitable for publication.

I have read through the revised version of the article and list below a number of points that require clarification prior to publication – please get in touch if anything is unclear. The points are all minor and therefore I recommend this article can now be 'published subject to technical corrections' (i.e. it does not need to undergo any further review by the editor or reviewers).

Thank you for choosing to publish your work in The Cryosphere.

Kind regards,

Pippa Whitehouse (editor)

**Minor line-by-line comments**

Line 20: "use of six different GHFs" – not clear if this refers to six different GHF models, or simply six different GHF values, please clarify and review use of this phrase throughout the manuscript
Reply: We change it to "use of six different GHF maps".

Line 32: does "subglacial rifts" refer to a geological feature or a glaciological feature? If the former, it is not clear how this is relevant when discussing the stability of the Lambert-Amery system on a decadal timescale – please clarify what type of feature you are referring to and how it is relevant
Reply: We refer it to a feature of the ice base which are defined by the bedrock, and change it to the term "subglacial canyons". They are relevant as presumably being similar features to the subglacial lakes that are known to empty and fill on multi-year timescales. We modified this part to "However, there is also evidence of extensive subglacial canyons and lakes (Fretwell et al., 2013; Jamieson et al., 2016; Cui et al., 2020a). Subglacial canyons and lakes are conduits for subglacial water, transporting subglacial meltwater to the coast through complex hydrologic routing, that may change

on relatively fast timescales (Malczyk et al., 2020). Jamieson et al. (2016) report a large subglacial drainage network in Princess Elizabeth Land (PEL), which would transport water from central PEL to the coast passing the Lambert-Amery region. Subglacial water can affect the ice flow (Stearns et al., 2008; Diez et al., 2018), influence the dynamical stability and basal mass balance (Gudlaugsson et al., 2017), and may enhance basal melt of ice shelves (Le Brocq et al., 2013)."

We add some references:

Malczyk, G., Gourmelen, N., Goldberg, D., Wuite, J., & Nagler, T.: Repeat subglacial lake drainage and filling beneath Thwaites Glacier. Geophys. Res. Lett., 47, e2020GL089658. https://doi.org/10.1029/2020GL089658, 2020.

Stearns, L. A., Smith, B. E., and Hamilton, G. S.: Increased flow speed on a large East Antarctic outlet glacier caused by subglacial floods, Nat. Geosci., 1, 827, 2008.

Diez, A., Matsuoka, K., Ferraccioli, F., Jordan, T. A., Corr, H. F., Kohler, J., Olesen, A. V., and Forsberg, R.: Basal Setings Control Fast Ice Flow in the Recovery/Slessor/Bailey Region, East Antarctica, Geophys. Res. Lett., 45, 2706–2715, 2018.

Gudlaugsson, E., Humbert, A., Andreassen, K., Clason, C. C., Kleiner, T., and Beyer, S.: Eurasian ice-sheet dynamics and sensitivity to subglacial hydrology, J. Glaciol., 63, 556–564, 2017.

Le Brocq, A. M., Ross, N., Griggs, J. A., Bingham, R. G., Corr, H. F. J., Ferraccioli, F., Jenkins, A., Jordan, T. A., Payne, A. J.,Rippin, D. M., Siegert, M. J., 2013, Evidence from ice shelves for channelized meltwater flow beneath the Antarctic Ice Sheet: Nat. Geosci., 6, 945–948, 2013.

Line 79: "…the basal thermal conditions inferred from the new high-resolution topography dataset" – include a reference to clarify which study you are referring to
Reply: Done.

Line 105: suggest "The margins of the inland sub-basins…"
Reply: We change it as you suggested.

Figure 1: it would be useful to depict the coastline/grounding line of Antarctica in plots (a)-(c). Also, I suggest adding text labels to plot (b) to clarify which domain uses data from Cui et al. (2020) and which domain uses data from MEaSUREs – one reviewer comments that this is still unclear
Reply: We improve Fig. 1 as suggested.

Line 165: clarify what you mean by a 'proper' initial temperature
Reply: We change "A proper initial ice temperature" to "A proper initial vertical ice temperature profile subject to thermal boundary conditions".

Line 170: clarify which models you are referring to, e.g. "coupling the forward and inverse models"

Reply: We change it to "coupling the forward and inverse models".

Line 179: "downhill in the ice surface" – unusual phrasing, perhaps 'along flowlines'?
Reply: Downhill is what we mean here, which is not always along flowlines. The term was used already in Wolovick et al., 2021. We changed the phrase to "downhill along the ice surface".

Line 185: 'step' -> 'component' (using the terminology from line 178)
Reply: The step (1) here refers to line 336-338. We remove "for step (1)" here to avoid confusion.

Equation 1: define 'm' and k(T), perhaps also stating whether they take positive or negative values
Reply: Done. We add "$k(T)$ is the temperature-dependent thermal conductivity of ice, $m$ is the basal melt rate."
Fig. 3 caption: specify (here and elsewhere) whether this is surface velocity or the full velocity field
Reply: Done. It is surface velocity.

Line 227: in what way is the basal slip ratio 'added' to the method? If the method already uses a basal slip ratio perhaps the novel feature here is that you use a spatially variable basal slip ratio?
Reply: Basal slip ratio is not used in the shape function in Wolovick et al. (2021a). So we simply mean we improve the method. We change the phrase to "We also improve the shape function in Wolovick et al. (2021a) by including the basal slip ratio"

Equations 3 and 4: suggest using the del/nabla symbol when representing div, grad etc.
Reply: Done.

Line 287: on line 275 you state that beta is the basal friction coefficient, check use of terminology
Reply: We remove the "$C$" in line 287.

Line 299: here and elsewhere, you could replace 'do', 'done', 'did' with 'carry out' or 'carried out' when referring to the methods used, e.g. "An L-curve analysis has been carried out to find…"
Reply: Thanks. We changed it to "An L-curve analysis has been carried out…".
We also change "Radar surveys have not yet been done…" in line 466 to "Radar surveys have not yet been carried out…"

Line 312: "…from by…" – typo
Reply: corrected.

Line 312: is TM the same as Tm (defined on line 202)?

Reply: Yes. We change $T_M$ to $T_m$.

Figure 5: state that this figure shows results for the Martos et al. (2017) model
Reply: Done.

Lines 331-332: could relate these statements about heat to the impact of each term on the basal melt rate – this would help to clarify the sign of the final term in the numerator of eq. 10
Reply: We change the sentence to "Geothermal heat and frictional heating from basal slip warm the base, while the upward heat conduction to the interior cools the base. Note that basal melt rate can be either positive (melting) or negative (freezing) depending on the heat balance."

Line 335: similar as -> similar to
Reply: done.

Line 336: refer to figure 3 when referring to the 'mixture' of information used to determine the input surface velocity (also on line 345 when discussing merging three different pieces of information)
Reply: Done.

Line 347: by -> from
Reply: Done.

Line 438: figure 9?
Reply: Yes. It should be Fig. 9. We corrected it.

Line 442-443: suggest "Regions with positive basal melt rates…"
Reply: Sorry we do not understand the point here. the Line 442-3 reads "where frictional heat is high (Fig. 5b), despite the differences in GHF (Fig. 2). Basal melt rate is above 10 mm yr$^{-1}$ near the grounding line, reaching 500 mm yr$^{-1}$ at the" and we do not see how to or why this needs changing as suggested.

Line 454-455: suggest "…the second greatest number of observed subglacial lakes…"
Reply: Done.

Line 477: suggest "the …accumulation rate field used in our modelling will be…" (similar on line 181)
Reply: Done. We think you mean line 481 rather than 181. We also made similar change on line 481.

Line 478: "because of lower accumulation rates during glacial periods" – reference needed
Reply: We add two references:

Watanabe, O., Shoji, H., Satow, K., Motoyama, H., Fujii, Y., Narita, H., and Aoki, S.: Dating of the Dome Fuji Antarctica deep ice core, Mem. Natl. Inst. Polar Res. Spec. Iss., 57, 25–37, 2003.

Van Ommen, T. D., Morgan, V., Curran, M. A. J., Deglacial and Holocene changes in accumulation at Law Dome, East Antarctic, Annals of Glaciology, 39, 359-365, 2004.

Line 507: suggest "…that does not predict basal melt at the location of the observed lakes…"
Reply: Done.

Line 509: a third possibility is that lakes are present, but we do not have the data to detect them
Reply: Yes, we add this possibility.

Line 515: "direct measurements… are rare" – the important point here is that lakes can exist where basal temperatures are below the pressure melting point, suggest revising the text to reflect this.
Reply: The point that lakes can exist where basal temperatures are below the pressure melting point is mentioned in line 512-513. Therefore, we remove "direct measurements… are rare". And change line 514 to "and no similar ones are known to exist beneath the Antarctic ice"

Line 517: the paper suggested by the reviewer 1 is by Tulaczyk et al. (2020) not Talalay et al. (2020) –see https://tc.copernicus.org/articles/14/4495/2020/
Reply: We changed the reference to Tulaczyk et al. (2020), and slightly modified the sentence to "Furthermore, relatively high electrical conductivity beds such as clay-rich sediments surrounded by bedrock can give rise to false positives in radar detections of subglacial water bodies (Tulaczyk et al., 2020)."

Line 530: Antarctica -> Antarctic
Reply: Done.

Line 533: "…to the our…" – typo
Reply: Done. We removed "the".

Line 557: include full references for the Martos and Li GHF models (also lines 592-594)
Reply: Done.

line 600: "…has smaller modelled basal friction coefficients…" – is this information quantified anywhere in the manuscript?
Reply: It is not quantified in the manuscript. We remove this half sentence, and change the whole sentence to "The fast-flowing region has fast basal velocities and high frictional heat, but there are large differences in basal melting rates between the 6 GHF

datasets.".

Please carry out a careful check for grammatical issues, particularly in relation to the use of singular and plural and the definite article ('the'), which is occasionally missing.